# Channel Normalization for Time Series Channel Identification

Seunghan Lee [1 2]   Taeyoung Park [* 1]   Kibok Lee [* 1]

## Abstract

Channel identifiability (CID) refers to the ability to distinguish among individual channels in time series (TS) modeling. The absence of CID often results in producing identical outputs for identical inputs, disregarding channel-specific characteristics. In this paper, we highlight the importance of CID and propose *Channel Normalization* (CN), a simple yet effective normalization strategy that enhances CID by assigning *distinct* affine transformation parameters to *each channel*. We further extend CN in two ways: 1) *Adaptive CN* (ACN) dynamically adjusts parameters based on the input TS, improving adaptability in TS models, and 2) *Prototypical CN* (PCN) introduces a set of learnable prototypes instead of per-channel parameters, enabling applicability to datasets with unknown or varying number of channels and facilitating use in TS foundation models. We demonstrate the effectiveness of CN and its variants by applying them to various TS models, achieving significant performance gains for both non-CID and CID models. In addition, we analyze the success of our approach from an information theory perspective. Code is available at https://github.com/seunghan96/CN.

## 1. Introduction

Time series (TS) forecasting is widely used in various fields, including traffic (Cirstea et al., 2022), electricity (Dudek et al., 2021), and sales forecasting (Li et al., 2022). A range of TS forecasting methods have been developed based on different architectures, such as Transformers (Vaswani et al., 2017), multi-layer perceptrons (MLPs) (Rumelhart et al., 1986), and state-space models (SSMs) (Gu & Dao, 2023). Among them, some models are inherently able to distinguish among channels (i.e., *channel-identifiable* or

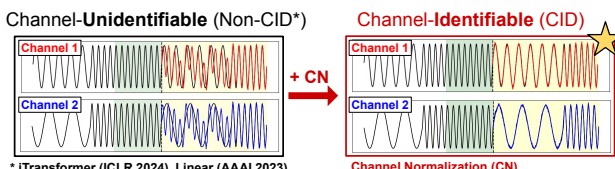

Figure 1: **Motivating example for channel identifiability.** When two different channels receive the locally identical inputs (green), a non-CID model yields the same outputs (yellow) for both, failing to distinguish between them, as shown in the left panel. In contrast, applying CN enables CID and produces distinct outputs even with the same inputs, as shown in the right panel.

| Average MSE ($4Hs$) | ETTm1 | Weather | PEMS03 | Imp. |
|---|---|---|---|---|
| iTransformer | .408 | .260 | .142 | - |
| + Constant vector | **.397** | **.246** | **.114** | **6.5%** |

Table 1: **Necessity of CID.** Simply adding different constant vectors to each channel token improves the performance. Full results and comparison with our methods are shown in Appendix G.

*CID*), while others are not (i.e., *channel-unidentifiable* or *non-CID*), producing identical outputs for the identical input regardless of the channel (Liu et al., 2024; Zeng et al., 2023).

Figure 1 illustrates the TS forecasting results using iTransformer (Liu et al., 2024), a widely adopted non-CID model, on a toy dataset with two channels displaying distinct patterns. The figure shows that the model fails on this simple task, as non-CID models lack information about channel identities, producing *identical* outputs (yellow) for both channels whenever given *identical* inputs (green). Furthermore, Table 1 shows that adding distinct constant vectors to each channel token, enabling the model to distinguish among channels, improves the forecasting performance. These results highlight the importance of CID in TS models.

A naive approach to solving this issue is to use different parameters for each channel in the tokenization layer, although this increases computational burden (Nie et al., 2024), or to add learnable vectors to each channel token (i.e., channel identifiers) (Chi et al., 2024). These methods yield limited performance gains, as discussed in Section 5.3, motivating us to design a simple yet effective method to enhance CID.

To this end, we propose *Channel Normalization* (CN), a simple yet effective normalization strategy designed to enhance CID of TS models. Unlike Layer Normalization (LN) (Ba et al., 2016) which applies *shared* affine transformation parameters across all channels, CN employs *distinct* parameters for *each channel*, allowing the model to differentiate

---

[*]Equal advising [1]Department of Statistics and Data Science, Yonsei University [2]KRAFTON; work done while at Yonsei University. Correspondence to: Taeyoung Park <tpark@yonsei.ac.kr>, Kibok Lee <kibok@yonsei.ac.kr>.

*Proceedings of the 42nd International Conference on Machine Learning*, Vancouver, Canada. PMLR 267, 2025. Copyright 2025 by the author(s).

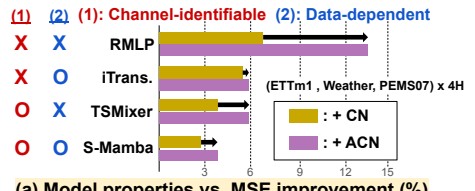 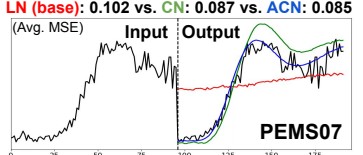 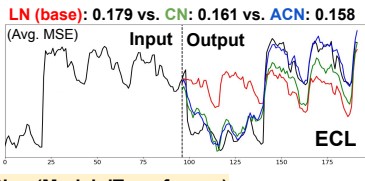

Figure 2: **Effectiveness of CN/ACN.** (a) shows that our method is effective across various backbones, where 1) *non-CID models* (e.g., RMLP, iTransformer) exhibit greater improvements from CN and 2) *data-independent models* (e.g., RMLP, TSMixer), whose parameters do not depend on the input, benefit more from transitioning from CN to ACN. (b) shows forecasting results with and without our methods.

among channels effectively. Furthermore, we introduce two variants of CN: 1) *Adaptive CN* (ACN), which dynamically adjusts parameters based on the input TS to improve adaptability, and 2) *Prototypical CN* (PCN), which introduces a set of learnable prototypes as affine transformation parameters after normalization to handle multiple datasets with unknown/varying number of channels using a single model, particularly useful for TS foundation models (TSFMs).

The main contributions are summarized as follows:

- We propose **CN** to enhance CID of TS models by employing channel-specific parameters, unlike Layer Normalization which uses shared parameters, offering a simple and effective strategy.
- We propose two variants of CN: 1) **ACN** to better capture time-varying characteristics of each channel by adapting its parameters to input TS and 2) **PCN** to handle multiple datasets with unknown/varying number of channels by introducing learnable prototypes where parameters are assigned to prototypes instead of channels.
- We provide extensive experiments on various backbones including TSFMs, achieving significant improvements for both CID and non-CID models as shown in Figure 2(a).
- We analyze the effect of our method from an information theory perspective, showing that it 1) enriches feature representations, 2) improves the uniqueness of each channel representation, and 3) diversifies the correlation between channel representations, supporting the performance gain.

## 2. Related Works

**TS forecasting models.** TS forecasting in deep learning has been approached with two strategies: channel-dependent (CD) strategy, which captures dependencies among channels, and channel-independent (CI) strategy, which treats each channel individually and focuses only on the temporal dependency (TD). Methods using these strategies include Transformer-based, MLP-based, and SSM-based models.

For Transformer-based models, PatchTST (Nie et al., 2023) divides TS into patches and feeds them into a Transformer in a CI manner. iTransformer (Liu et al., 2024) treats each channel as a token to capture CD using the attention mechanism, resulting in significant performance gains. However, these models suffer from the quadratic complexity of the attention mechanism. To overcome this issue, various MLP-based models have been proposed, where DLinear (Zeng et al., 2023) uses a linear model to capture TD, RLinear and RMLP (Li et al., 2023) integrate reversible normalization (RevIN) (Kim et al., 2021) to MLPs, and TSMixer (Chen et al., 2023) adopts MLPs to capture both TD and CD. Recently, various methods (Ahamed & Cheng, 2024; Ma et al., 2024; Zeng et al., 2024; Cai et al., 2024) have been proposed that utilize Mamba (Gu & Dao, 2023), which introduces a selective scan mechanism to SSM to capture long-range context with linear complexity. S-Mamba (Wang et al., 2025) and Bi-Mamba+ (Liang et al., 2024) capture CD with bidirectional Mamba, and SOR-Mamba (Lee et al., 2024) employs a regularization strategy to effectively capture CD.

Recently, several methods have emerged that enhance CID of TS models. InjectTST (Chi et al., 2024) proposes a channel identifier that helps a Transformer differentiate among channels and C-LoRA (Nie et al., 2024) conditions a CD model on channel-specific components using a channel-aware low-rank adaptation method. Similarly, CCM (Chen et al., 2024a) integrates channel-cluster identity to a TS model by grouping channels based on their similarities. However, these methods, aside from the channel identifier, were not primarily developed to enhance CID, and their impact on CID is merely a byproduct. Furthermore, they either require modifications to the architecture (Chen et al., 2024a; Nie et al., 2024) or provide only limited performance gains (Chi et al., 2024; Nie et al., 2024), as shown in Table 6.

**Normalization.** Various normalization methods for deep neural networks have been introduced (Ioffe, 2015; Wu & He, 2018) to improve convergence and training stability, differing in the dimension they normalize. Layer Normalization (LN) (Ba et al., 2016), which uses *shared* affine transformation parameters across channels, is commonly employed in TS backbones (Nie et al., 2023; Wang et al., 2025) to reduce inter-channel discrepancies (Liu et al., 2024). In contrast to LN, we propose assigning *channel-specific* parameters to distinguish among channels.

## 3. Preliminaries

**TS forecasting (TSF).** In TSF tasks, a model predicts the future values $\mathbf{y} = (\mathbf{x}_{L+1}, \ldots, \mathbf{x}_{L+H})$ with a lookback window (i.e., input TS) $\mathbf{x} = (\mathbf{x}_1, \ldots, \mathbf{x}_L)$. In this setup, each $\mathbf{x}_i \in \mathbb{R}^C$ represents values at individual time steps, with $L$, $H$, and $C$ indicating the size of the lookback window, the forecast horizon, and the number of channels, respectively.

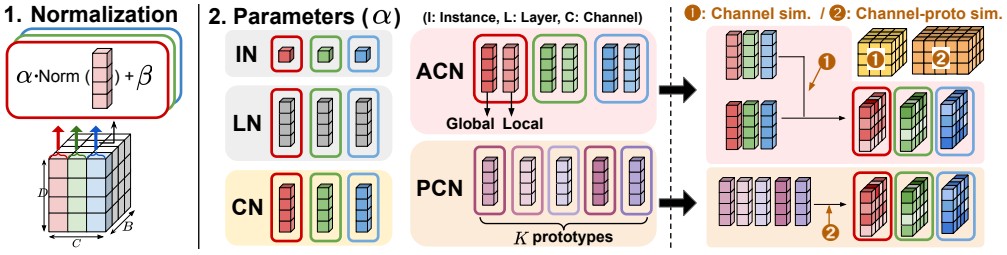

Figure 3: **Overall framework of CN/ACN/PCN.** (1) **CN** employs *channel-specific* parameters, enabling the model to distinguish among channels. (2) **ACN** extends CN by adapting its parameters to the input TS by utilizing *local parameters*, which are attended to with different weights based on the similarity between input channels (i.e., *channel similarity*). (3) **PCN** makes CN applicable to multiple datasets with unknown/varying number of channels by assigning parameters to each *prototype* instead of each channel, which are attended to with different weights depending on the similarity between input channels and prototypes (i.e., *channel-prototype similarity*).

**Framework of TSF models.** General TS forecasting models follow the framework below:

$$\text{(Optional): } \mathbf{x} \leftarrow \text{NORMALIZE}(\mathbf{x})$$
$$\text{1. Token Embedding: } \mathbf{z} \leftarrow g_1(\mathbf{x}),$$
$$\text{2. Encoder: } \mathbf{z} \leftarrow f(\mathbf{z}), \qquad (1)$$
$$\text{3. Projection Layer: } \hat{\mathbf{y}} \leftarrow g_2(\mathbf{z}),$$
$$\text{(Optional): } \hat{\mathbf{y}} \leftarrow \text{DENORMALIZE}(\hat{\mathbf{y}}),$$

where $\mathbf{z}$ is a $D$-dimensional vector and various normalization methods apply within $f$ for training stability (e.g., LN). Similarly, our method applies within $f$, remaining orthogonal to techniques that normalize the input ($\mathbf{x}$) and denormalize the output ($\hat{\mathbf{y}}$) to address distribution shifts (Passalis et al., 2019; Kim et al., 2021).

**Model property 1: Channel identifiability.** Let $X = \{x_1, \ldots, x_C\} \in \mathbb{R}^{L \times C}$ be an input TS with $C$ channels of length $L$. A TSF model $\phi$ exhibits CID if it can distinguish among channels with identical input. That is, A CID model can produce distinct outputs $\phi(X)_i \neq \phi(X)_j$ even with the same inputs $x_i = x_j$, whereas a non-CID model produces the same outputs $\phi(X)_i = \phi(X)_j$.

**Model property 2: Data dependency.** A TSF model $\phi$ is *data dependent* (Chen et al., 2023) if its parameters adapt to the input TS (e.g., attention in Transformers or selective scanning in Mamba). In contrast, $\phi$ is *data independent* if its parameters are fixed across the input TS (e.g., linear models). Data-dependent models exhibit high representational capacity, whereas data-independent models are simpler and less prone to overfitting (Chen et al., 2023).

## 4. Methodology

In this section, we introduce CN[1] which employs channel-specific parameters, unlike LN which employs shared parameters, to enhance CID of TS models. Furthermore, we propose two variants: ACN, which adjusts the parameters based on the input TS, and PCN, which handles multiple datasets with unknown or varying number of channels. The overall framework is shown in Figure 3.

---

[1]CN can serve as both a strategy (framework) and a method.

### 4.1. Channel Normalization (CN)

**Layer Normalization (LN).** LN applies affine transformations with parameters $\{\alpha, \beta\}$ to the normalized data as:

$$\text{Norm}(z_{b,c,d}) = \frac{z_{b,c,d} - \mu_{b,c}}{\sigma_{b,c}}, \qquad (2)$$
$$\hat{z}_{b,c,d} = \alpha \cdot \text{Norm}(z_{b,c,d}) + \beta.$$

Various TS methods apply LN by using shared parameters $\{\alpha, \beta\}$ across channels to reduce discrepancies among the channels (Liu et al., 2024; Wang et al., 2025).

**Channel Normalization (CN).** Unlike LN, which uses shared affine transformation parameters across channels, CN employs channel-specific parameters as follows:

$$\hat{z}_{b,c,d} = \alpha_c \cdot \text{Norm}(z_{b,c,d}) + \beta_c, \qquad (3)$$

where $\alpha_c$ and $\beta_c$ denotes the parameters of the $c$-th channel. This simple modification enables the model to distinguish among channels, with the additional computational burden of using channel-specific parameters being minor compared to the shared parameters of LN, as shown in Table 8.

**Variants of CN.** To further enhance the flexibility and applicability of CN, we propose two variants, ACN and PCN, addressing the following questions, respectively:
- Q1) As the parameters of CN are independent on input and unable to capture the dynamic characteristics of each input channel, how can we make them adapt to the input?
- Q2) As CN requires a predefined number of channels, how can we handle *multiple* datasets with *unknown or varying* number of channels (e.g., training TSFMs)?

### 4.2. Adaptive Channel Normalization (ACN)

The parameters of CN are fixed across time steps and independent of the input TS. However, the characteristics of each channel may vary over time due to distribution shifts (Han et al., 2023). To this end, we propose ACN by introducing *local* parameters ($\alpha_c^L$) to CN, which are attended to with different weights depending on the input channels. To distinguish local parameters from the original parameters of CN, we refer to the original parameters as *global* parameters ($\alpha_c^G$), as they are shared globally across the time steps.

**Algorithm 1** Channel Normalization (CN)

**Require:**
1: Input $z \in \mathbb{R}^{B \times C \times D}$
2: Parameters $\alpha, \beta \in \mathbb{R}^{C \times D}$
**Ensure:** Output $\hat{z} \in \mathbb{R}^{B \times C \times D}$
3: **for** $b = 1, \ldots, B$ **do**
4:    **for** $c = 1, \ldots, C$ **do**
5:       **for** $d = 1, \ldots, D$ **do**
6:          $\hat{z}_{b,c,d} = \alpha_{c,d} \cdot \mathrm{Norm}(z_{b,c,d}) + \beta_{c,d}$
7:       **end for**
8:    **end for**
9: **end for**

**Channel similarity.** To attend to local parameters adaptively based on the input, we construct a *channel similarity* matrix $\hat{S} \in \mathbb{R}^{B \times C \times C}$, representing the similarity between the input channels, where $B$ denotes a batch size. Specifically, we use cosine similarity, which is then normalized by softmax with temperature $\tau$ as:

$$S_{b,c_1,c_2} = \frac{z_{b,c_1} \cdot z_{b,c_2}}{\|z_{b,c_1}\| \, \|z_{b,c_2}\|}, \tag{4}$$

$$\hat{S}_{b,c_1,c_2} = \frac{\exp\left(S_{b,c_1,c_2}/\tau\right)}{\sum_{i=1}^{C} \exp\left(S_{b,c_1,i}/\tau\right)}, \tag{5}$$

where $b \in \{1, \ldots, B\}$ and $c_1, c_2 \in \{1, \ldots, C\}$. This matrix $S$ serves as dynamic weights to obtain (dynamic) parameter $\hat{\alpha}_{b,c}^{\mathrm{L}} \in \mathbb{R}^{D}$ from the (static) parameter $\alpha_c^{\mathrm{L}} \in \mathbb{R}^{D}$ as below[2]:

$$\hat{\alpha}_{b,c}^{\mathrm{L}} = \sum_{i=1}^{C} \hat{S}_{b,c,i} \cdot \alpha_i^{\mathrm{L}}, \tag{6}$$

where $\hat{S}_{b,c,i}$ is the similarity between the $c$-th and the $i$-th channel of the $b$-th data, $\alpha_i^{\mathrm{L}}$ is the (static) local parameter of the $i$-th channel, and $\hat{\alpha}_{b,c}^{\mathrm{L}}$ is the resulting (dynamic) local parameters of the $c$-th channel of the $b$-th data, representing the weighted average of $\alpha_i^{\mathrm{L}}$ using $\hat{S}_{b,c,i}$ as dynamic weights.

The parameters of ACN are constructed by element-wise multiplication of the global and dynamic local parameters ($\alpha_c^{\mathrm{G}} \circ \hat{\alpha}_{b,c}^{\mathrm{L}}$), which complement each other, as shown in Table 7. Further analyses regarding the robustness to the similarity metric, $\tau$, and the space where the similarity is calculated are shown in Appendix H, J, and K, respectively.

### 4.3. Prototypical Channel Normalization (PCN)

Since CN assign parameters to *each channel*, it is infeasible to handle datasets with an unknown $C$ (e.g., inference on unseen datasets) or to train on multiple datasets with varying $C$s (e.g., require parameters for all channels in all datasets). To address this issue, we propose PCN by introducing learnable prototypes, where learnable parameters are assigned to *each prototype* instead of *each channel*, enabling it to handle an arbitrary number of channels. Similar to ACN, these prototype parameters ($\alpha_k^{\mathrm{P}}$) are attended to with different weights depending on the input TS.

---
[2]The same procedure is applied to $\beta$ as to $\alpha$.

**Algorithm 2** Adaptive Channel Normalization (ACN)

**Require:**
1: Input $z \in \mathbb{R}^{B \times C \times D}$
2: Channel similarity matrix $\hat{S} \in \mathbb{R}^{B \times C \times C}$
3: Global and local parameters $\alpha^{\mathrm{G}}, \alpha^{\mathrm{L}}, \beta^{\mathrm{G}}, \beta^{\mathrm{L}} \in \mathbb{R}^{C \times D}$
**Ensure:** Output $\hat{z} \in \mathbb{R}^{B \times C \times D}$
4: **for** $b = 1, \ldots, B$ **do**
5:    **for** $c = 1, \ldots, C$ **do**
6:       **for** $d = 1, \ldots, D$ **do**
7:          $\alpha_{b,c,d} = \alpha_{c,d}^{\mathrm{G}} \cdot (\sum_{i=1}^{C} \hat{S}_{b,c,i} \cdot \alpha_{i,d}^{\mathrm{L}})$
8:          $\beta_{b,c,d} = \beta_{c,d}^{\mathrm{G}} \cdot (\sum_{i=1}^{C} \hat{S}_{b,c,i} \cdot \beta_{i,d}^{\mathrm{L}})$
9:          $\hat{z}_{b,c,d} = \alpha_{b,c,d} \cdot \mathrm{Norm}(z_{b,c,d}) + \beta_{b,c,d}$
10:       **end for**
11:    **end for**
12: **end for**

---

**Algorithm 3** Prototypical Channel Normalization (PCN)

**Require:**
1: Input $z \in \mathbb{R}^{B \times C \times D}$
2: Channel-proto similarity matrix $\hat{S}^{\alpha}, \hat{S}^{\beta} \in \mathbb{R}^{B \times C \times K}$
3: Prototype parameters $\alpha^{\mathrm{P}}, \beta^{\mathrm{P}} \in \mathbb{R}^{K \times D}$
**Ensure:** Output $\hat{z} \in \mathbb{R}^{B \times C \times D}$
4: **for** $b = 1, \ldots, B$ **do**
5:    **for** $c = 1, \ldots, C$ **do**
6:       **for** $d = 1, \ldots, D$ **do**
7:          $\alpha_{b,c,d} = \sum_{i=1}^{K} \hat{S}_{b,c,i}^{\alpha} \cdot \alpha_{i,d}^{\mathrm{P}}$
8:          $\beta_{b,c,d} = \sum_{i=1}^{K} \hat{S}_{b,c,i}^{\beta} \cdot \beta_{i,d}^{\mathrm{P}}$
9:          $\hat{z}_{b,c,d} = \alpha_{b,c,d} \cdot \mathrm{Norm}(z_{b,c,d}) + \beta_{b,c,d}$
10:       **end for**
11:    **end for**
12: **end for**

**Channel-prototype similarity.** To enable channels with an arbitrary number to utilize the prototype parameters, we construct a *channel-prototype similarity* matrix $\hat{S}^{\alpha} \in \mathbb{R}^{B \times C \times K}$, representing the similarity between input channels and prototypes. Note that rather than employing a latent space ($z$) to represent channels, we apply an additional projection layer ($h$) in the data space ($x$) to align with the prototype space. Specifically, we use cosine similarity, which is then normalized by softmax with temperature $\tau$ as:

$$S_{b,c,k}^{\alpha} = \frac{h(x_{b,c}) \cdot \alpha_k^{\mathrm{P}}}{\|h(x_{b,c})\| \, \|\alpha_k^{\mathrm{P}}\|}, \tag{7}$$

$$\hat{S}_{b,c,k}^{\alpha} = \frac{\exp\left(S_{b,c,k}^{\alpha}/\tau\right)}{\sum_{i=1}^{K} \exp\left(S_{b,c,i}^{\alpha}/\tau\right)}, \tag{8}$$

where $k \in \{1, \ldots, K\}$, $K$ is the number of prototypes, and $h$ is a linear projection layer. Similar to ACN, this matrix is used as dynamic weights to obtain $\hat{\alpha}_{b,c}^{\mathrm{P}}$ from $\alpha_k$ as below:

$$\hat{\alpha}_{b,c}^{\mathrm{P}} = \sum_{i=1}^{K} \hat{S}_{b,c,i}^{\alpha} \cdot \alpha_i^{\mathrm{P}}. \tag{9}$$

Further analyses of the robustness to $K$ and the employment of $h$ are demonstrated in Appendix I and K.2, respectively.

| Non-CID models | iTransformer | | + CN | | + ACN | | Imp. (MSE) | | RMLP | | + CN | | + ACN | | Imp. (MSE) | |
|---|---|---|---|---|---|---|---|---|---|---|---|---|---|---|---|---|
| Datasets | MSE | MAE | MSE | MAE | MSE | MAE | + CN | + ACN | MSE | MAE | MSE | MAE | MSE | MAE | + CN | + ACN |
| ETTh1 | .457 | .449 | .441 | .439 | .438 | .438 | 3.5% | 4.2% | .471 | .453 | .445 | .437 | .448 | .435 | 5.5% | 4.9% |
| ETTh2 | .384 | .407 | .376 | .404 | .374 | .402 | 2.1% | 2.6% | .381 | .408 | .380 | .405 | .376 | .402 | 0.3% | 1.3% |
| ETTm1 | .408 | .412 | .396 | .403 | .395 | .402 | 2.9% | 3.2% | .401 | .406 | .384 | .397 | .383 | .396 | 4.2% | 4.5% |
| ETTm2 | .293 | .337 | .289 | .331 | .288 | .330 | 1.4% | 1.7% | .280 | .326 | .277 | .324 | .277 | .323 | 1.1% | 1.1% |
| PEMS03 | .142 | .248 | .101 | .204 | .098 | .203 | 31.0% | 38.0% | .205 | .294 | .192 | .284 | .159 | .266 | 6.3% | 22.4% |
| PEMS04 | .121 | .232 | .088 | .196 | .088 | .195 | 27.3% | 27.3% | .236 | .321 | .212 | .304 | .156 | .265 | 10.2% | 33.9% |
| PEMS07 | .102 | .205 | .087 | .178 | .085 | .174 | 14.7% | 16.7% | .200 | .284 | .184 | .270 | .131 | .233 | 8.0% | 34.5% |
| PEMS08 | .254 | .306 | .159 | .223 | .153 | .221 | 37.4% | 39.8% | .277 | .333 | .247 | .308 | .187 | .279 | 10.8% | 32.5% |
| Exchange | .368 | .409 | .352 | .401 | .349 | .398 | 4.4% | 5.2% | .356 | .403 | .355 | .400 | .353 | .399 | 0.3% | 0.8% |
| Weather | .260 | .281 | .247 | .273 | .245 | .271 | 5.0% | 5.8% | .272 | .292 | .249 | .274 | .246 | .273 | 8.5% | 9.6% |
| Solar | .234 | .261 | .228 | .258 | .220 | .253 | 2.6% | 6.0% | .261 | .313 | .248 | .276 | .242 | .277 | 5.0% | 7.3% |
| ECL | .179 | .270 | .161 | .256 | .158 | .256 | 10.1% | 11.7% | .228 | .313 | .190 | .277 | .189 | .276 | 16.7% | 17.1% |
| Average | .275 | .318 | .244 | .297 | .241 | .295 | 11.3% | 12.4% | .297 | .346 | .280 | .330 | .262 | .319 | 5.7% | 11.8% |
| Best count (/48) | 0 | 0 | 9 | 9 | 46 | 46 | Δ Imp.: | 1.1%p | 0 | 0 | 4 | 7 | 44 | 46 | Δ Imp.: | 6.1%p |
| CID models | S-Mamba | | + CN | | + ACN | | Imp. (MSE) | | TSMixer | | + CN | | + ACN | | Imp. (MSE) | |
| ETTh1 | .457 | .452 | .455 | .450 | .448 | .446 | 0.4% | 2.0% | .462 | .449 | .438 | .435 | .453 | .441 | 5.2% | 1.9% |
| ETTh2 | .383 | .408 | .375 | .401 | .374 | .400 | 2.1% | 2.3% | .403 | .418 | .387 | .410 | .386 | .407 | 4.0% | 4.2% |
| ETTm1 | .398 | .407 | .397 | .406 | .394 | .404 | 0.3% | 1.0% | .401 | .406 | .386 | .398 | .385 | .397 | 3.7% | 4.0% |
| ETTm2 | .290 | .333 | .286 | .329 | .284 | .328 | 1.4% | 2.1% | .287 | .330 | .286 | .329 | .280 | .325 | 0.3% | 2.4% |
| PEMS03 | .133 | .240 | .108 | .214 | .107 | .213 | 18.8% | 19.5% | .129 | .236 | .124 | .228 | .120 | .230 | 3.9% | 7.0% |
| PEMS04 | .096 | .205 | .085 | .189 | .095 | .202 | 11.5% | 1.0% | .115 | .228 | .114 | .222 | .109 | .222 | 0.9% | 5.2% |
| PEMS07 | .090 | .191 | .078 | .168 | .073 | .167 | 13.3% | 18.9% | .115 | .210 | .115 | .209 | .103 | .203 | 0.0% | 10.4% |
| PEMS08 | .157 | .242 | .133 | .216 | .121 | .216 | 15.3% | 22.9% | .186 | .275 | .167 | .250 | .167 | .258 | 10.2% | 10.2% |
| Exchange | .364 | .407 | .362 | .405 | .357 | .402 | 0.5% | 1.9% | .365 | .406 | .358 | .402 | .356 | .400 | 1.9% | 2.5% |
| Weather | .252 | .277 | .246 | .273 | .247 | .274 | 2.4% | 2.0% | .260 | .285 | .246 | .274 | .242 | .272 | 5.4% | 6.9% |
| Solar | .244 | .275 | .230 | .262 | .228 | .261 | 5.7% | 6.6% | .255 | .294 | .246 | .267 | .245 | .274 | 3.5% | 3.9% |
| ECL | .174 | .269 | .163 | .261 | .162 | .259 | 6.3% | 6.9% | .211 | .310 | .181 | .280 | .174 | .273 | 14.2% | 17.8% |
| Average | .253 | .309 | .243 | .298 | .240 | .297 | 4.0% | 5.1% | .266 | .321 | .254 | .309 | .243 | .308 | 4.5% | 8.6% |
| Best count (/48) | 1 | 0 | 15 | 25 | 38 | 31 | Δ Imp.: | 1.1%p | 0 | 0 | 10 | 16 | 40 | 36 | Δ Imp.: | 4.1%p |

Table 2: **Results of TS forecasting.** We apply CN/ACN to non-CID and CID models, achieving performance gains across all models.

# 5. Experiments

**Experimental setups.** We demonstrate the effectiveness of our method on TSF tasks with 12 datasets. For evaluation metrics, we use mean squared error (MSE) and mean absolute error (MAE). We follow the experimental setups from C-LoRA (Nie et al., 2024), with size of the lookback window ($L$) set to 96, and divide all datasets into training, validation, and test sets in chronological order. Further details of the setups are provided in Appendix A.

**Datasets.** For the experiments, we use 12 datasets: four ETT datasets (ETTh1, ETTh2, ETTm1, ETTm2) (Zhou et al., 2021), four PEMS datasets (PEMS03, PEMS04, PEMS07, PEMS08) (Chen et al., 2001), Exchange, Weather, ECL (Wu et al., 2021), and Solar-Energy (Solar) (Lai et al., 2018). Details of the dataset statistics are provided in Appendix A.1.

**Backbones.** For the experiments, we select four backbones: iTransformer (Liu et al., 2024), RMLP (Li et al., 2023), S-Mamba (Wang et al., 2025), and TSMixer (Li et al., 2023). For backbones that utilize LN, we replace it with our method, and for those without, we add our method. As illustrated in Figure 2(a), these methods can be categorized based on their a) inherent *CID ability* and b) *data dependency* of model parameters. Furthermore, for PCN, we employ UniTS (Gao et al., 2024), a TSFM that addresses diverse tasks using prompt-tuning, to demonstrate its capability to handle multiple datasets with varying $C$s and perform inference on unseen datasets with unknown $C$s. The baseline results are obtained from previous works (Nie et al., 2024; Lee et al.,

2024) and replicated using the official codes.

## 5.1. Application of CN/ACN

**TS forecasting.** Table 2 presents the average performance across four horizons ($H \in \{96, 192, 336, 720\}$), demonstrating that both CN and ACN consistently improve across all datasets and backbones, with ACN yielding additional gains compared to CN. Below, we analyze the performance gain from CN and the additional gain from ACN in relation to the two properties of the backbones.

**a) CID vs. non-CID.** As CN enhances the CID of models, it provides substantial improvements for *non-CID* models (e.g., iTransformer, RMLP), as shown in Figure 2(a). However, it also benefits *CID* models (e.g., S-Mamba, TSMixer) that *already* have the ability to distinguish among channels, although the improvements are relatively smaller. This is further validated by the results in Table 2, where non-CID models exhibit greater performance gains than CID models.

**b) Data dependent vs. Data independent.** As ACN improves upon CN by adapting to the input TS, transitioning from CN to ACN provides substantial improvements for *data-independent* models (e.g., RMLP, TSMixer), as shown in Figure 2(a). However, it also benefits *data-dependent* models (e.g., iTransformer, S-Mamba) whose parameters *already* adapt to the input TS, although the improvements are relatively smaller. This is further validated by the results in Table 2, where data-independent models exhibit greater additional performance gains from CN to ACN.

| ($N$: # datasets, $C_i$: # channels of $i$-th dataset) | | |
|---|---|---|
| | # Parameters | Zero-shot |
| CN | $2\sum_{i=1}^{N} C_i \cdot D$ | ✗ |
| ACN | $4\sum_{i=1}^{N} C_i \cdot D$ | ✗ |
| PCN | $2K \cdot D$ | ✓ |

Table 3: PCN for TSFMs.

| Metric (Best #) | | UniTS | + PCN | Imp. |
|---|---|---|---|---|
| 20 FCST (MSE) | Sup. | .469 (4) | **.433** (16) | **7.7%** |
| | Pmt. | .478 (3) | **.453** (20) | **5.2%** |
| 18 CLS (Acc.) | Sup. | 80.6 (2) | **83.0** (16) | **3.0%** |
| | Pmt. | 75.1 (3) | **79.5** (16) | **5.5%** |

Table 4: PCN to TSFMs.

| 12 Datasets | MSE | Imp. |
|---|---|---|
| iTransformer | .275 | - |
| + CN | .244 | 11.3% |
| + ACN | .241 | **12.4%** |
| + PCN | .252 | 8.4% |

Table 5: PCN to single-task models.

| | | Average MSE across 4 horizons | ETTh1 | ETTh2 | ETTm1 | ETTm2 | PEMS03 | PEMS04 | PEMS07 | PEMS08 | Exchange | Weather | Solar | ECL | Avg. | Imp. |
|---|---|---|---|---|---|---|---|---|---|---|---|---|---|---|---|---|
| Non-CID | iTransformer | - | .457 | .384 | .408 | .293 | .142 | .121 | .102 | .254 | .368 | .260 | .234 | .179 | .275 | - |
| | | + C-token | .450 | .389 | .400 | .290 | .123 | .109 | .106 | .157 | .376 | .246 | .255 | .169 | .256 | 6.9% |
| | | + C-project | .452 | .381 | .399 | **.286** | .119 | .109 | .097 | .163 | .366 | **.244** | .230 | .163 | .251 | 8.7% |
| | | + Channel identifier | .445 | .382 | .397 | .293 | .100 | .093 | **.082** | .168 | .365 | .248 | .231 | .165 | .248 | 9.8% |
| | | + C-LoRA | .450 | .392 | .398 | .289 | .114 | .113 | .106 | .169 | .364 | .248 | .241 | .167 | .254 | 7.6% |
| | | + ACN | **.438** | **.374** | **.395** | .288 | **.098** | **.088** | .085 | **.153** | **.349** | .245 | **.220** | **.158** | **.241** | **12.4%** |
| | RMLP | - | .471 | .381 | .401 | .280 | .205 | .236 | .200 | .277 | .356 | .272 | .261 | .228 | .297 | - |
| | | + C-token | .455 | .391 | .385 | **.277** | .220 | .218 | .196 | .286 | .368 | .246 | .267 | .205 | .293 | 1.4% |
| | | + C-project | .455 | .389 | .384 | **.277** | .186 | .190 | .172 | .233 | .366 | .245 | .249 | .195 | .278 | 6.3% |
| | | + Channel identifier | .452 | .380 | .393 | .279 | .191 | .209 | .185 | .262 | .356 | .250 | .254 | .199 | .284 | 4.4% |
| | | + C-LoRA | .451 | .379 | **.383** | .279 | .192 | .198 | .182 | .264 | .359 | **.245** | .256 | .190 | .282 | 5.1% |
| | | + ACN | **.448** | **.376** | **.383** | **.277** | **.159** | **.156** | **.131** | **.187** | **.353** | .246 | **.242** | **.189** | **.262** | **11.8%** |
| CID | S-Mamba | - | .457 | .383 | .398 | .290 | .133 | .096 | .090 | .157 | .364 | .252 | .244 | .174 | .253 | - |
| | | + C-token | .463 | .383 | .400 | **.285** | .117 | .087 | .097 | .134 | .376 | .245 | .244 | .171 | .250 | 1.2% |
| | | + C-project | .466 | .400 | .405 | .294 | .122 | .089 | .099 | .151 | .391 | .249 | .266 | .176 | .259 | -2.4% |
| | | + Channel identifier | .457 | .406 | .399 | .287 | .112 | .086 | .078 | .137 | .360 | .248 | .239 | .167 | .248 | 2.0% |
| | | + C-LoRA | .457 | .405 | .399 | .289 | .112 | **.084** | .092 | .144 | .359 | .247 | .238 | .169 | .250 | 1.2% |
| | | + ACN | **.448** | **.374** | **.394** | .284 | **.107** | .095 | **.073** | **.121** | .357 | .247 | **.228** | **.162** | **.240** | **5.1%** |
| | TSMixer | - | .462 | .403 | .401 | .287 | .129 | .115 | .115 | .186 | .365 | .260 | .255 | .211 | .266 | - |
| | | + C-token | .456 | .417 | .402 | .327 | .230 | .221 | .154 | .258 | .413 | .271 | .279 | .291 | .310 | -16.5% |
| | | + C-project | .457 | .412 | .401 | .324 | .232 | .222 | .154 | .268 | .407 | .271 | .275 | .281 | .309 | -16.2% |
| | | + Channel identifier | .454 | .390 | .394 | .284 | .124 | .114 | .106 | .185 | .355 | .245 | .251 | .186 | .257 | 2.3% |
| | | + C-LoRA | .460 | .407 | .399 | .283 | .122 | .110 | .103 | .181 | .366 | .245 | .251 | .187 | .260 | 3.4% |
| | | + ACN | **.453** | **.386** | **.385** | **.280** | **.120** | **.109** | **.103** | **.167** | **.356** | **.242** | **.245** | **.174** | **.243** | **8.6%** |

Table 6: **Comparison with other methods.** We compare ACN with 1) *baseline methods*, which employ channel-specific parameters for token embedding (**C-token**) or projection layers (**C-project**), and 2) *previous methods*, including **channel identifier** and **C-LoRA**.

## 5.2. Application of PCN

**Application to TSFMs.** As shown in Table 3, applying CN and ACN to TSFM is infeasible due to 1) the substantial increase in parameters, as it requires parameters for all channels across all datasets and 2) their inability to handle unseen datasets during training, as the number of channels may differ between training and inference datasets. In contrast, PCN addresses these limitations by employing prototypes. Table 4 presents the application of PCN to UniTS, showing the average results for 20 forecasting and 18 classification tasks under supervised (Sup.) and prompt-tuning (Pmt.) settings, with consistent improvements observed across all tasks. Full results of Table 4 and improvements on zero-shot forecasting tasks are provided in Appendix N and L.

**Application to single-task[3] models.** Although PCN is designed for TSFMs, it also improves the performance of single-task models trained on a single dataset, even when the number of channels is unknown. As shown in Table 5, applying PCN with $K = 5$ to iTransformer improves performance by 8.4% on average across 12 datasets and 4 horizons, though the improvement is smaller than that achieved by CN and ACN. We attribute this to the fact that, unlike CN and ACN which assign each *channel* a distinct parameter, PCN assigns each *prototype* (channel cluster) a distinct parameter,

---
[3]A single-task model is trained on a *single* dataset.

resulting in a weaker enforcement of CID.

## 5.3. Comparison with Other Methods

To demonstrate the effectiveness of ACN, we compare it with two categories of methods: 1) *baseline methods*, which use a naive strategy of employing channel-specific parameters for token embedding (**C-token**) or projection layers (**C-project**), and 2) *previous methods*, including **channel identifier** (Chi et al., 2024), a learnable vector added to channel tokens and **C-LoRA** (Nie et al., 2024), which applies a channel-aware LoRA to TS models. Table 6 demonstrates that our method outperforms these approaches across all backbones, while two baseline methods (C-token and C-project) even degrade the performance of CID models.

## 6. Analysis

In this section, we conduct **(1) ablation studies** on ACN, **(2) entropy analyses** to explain the proposed method from an information theory perspective, and **(3) other analyses** including both qualitative and quantitative evaluations.

## 6.1. Ablation Study

To demonstrate the effectiveness of ACN, we conduct an ablation study of using the global and local parameters with iTransformer. Table 7 presents the results, indicating that using all components yields the best performance.

| ACN | | Average MSE across 4 horizons | | | | | | | | | | | | Avg. | Imp. |
|---|---|---|---|---|---|---|---|---|---|---|---|---|---|---|---|
| Adaptive | CN | ETTh1 | ETTh2 | ETTm1 | ETTm2 | PEMS03 | PEMS04 | PEMS07 | PEMS08 | Exchange | Weather | Solar | ECL | | |
| | | .457 | .384 | .408 | .293 | .142 | .121 | .102 | .254 | .368 | .260 | .234 | .179 | .275 | - |
| ✓ | | .439 | .375 | **.395** | .289 | .108 | .110 | .099 | .174 | **.347** | .247 | .226 | .162 | .247 | 10.2% |
| | ✓ | .441 | .376 | .396 | .289 | .101 | **.088** | .087 | .159 | .352 | .247 | .228 | .161 | .244 | 11.3% |
| ✓ | ✓ | **.438** | **.374** | **.395** | **.288** | **.098** | **.088** | **.085** | **.153** | .349 | **.245** | **.220** | **.158** | **.241** | **12.4%** |

Table 7: **Ablation study.** None:$\{\alpha_d\}$ vs. Adaptive (local parameters):$\{\hat{\alpha}^{\mathrm{L}}_{b,c,d}\}$ vs. CN (global parameters):$\{\alpha^{\mathrm{G}}_{c,d}\}$ vs. ACN:$\{\alpha^{\mathrm{G}}_{c,d}\cdot\hat{\alpha}^{\mathrm{L}}_{b,c,d}\}$.

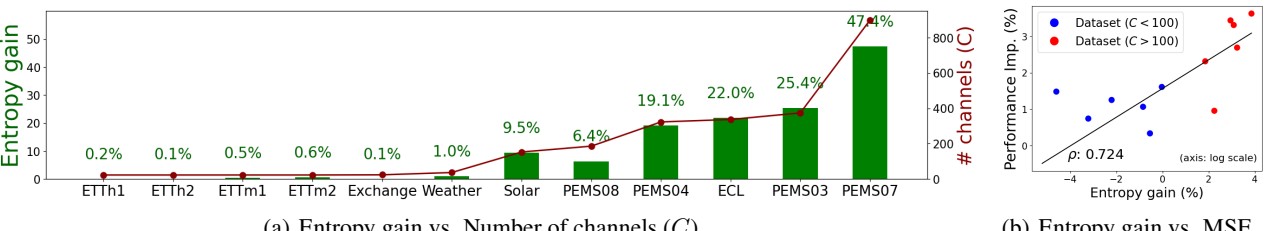

(a) Entropy gain vs. Number of channels ($C$).   (b) Entropy gain vs. MSE.

Figure 4: **Channel entropy gain by CN.** (a) Datasets with higher $C$ show a higher *entropy gain*. (b) Datasets with higher *entropy gain* show a higher *performance gain* (average MSE across four horizons).

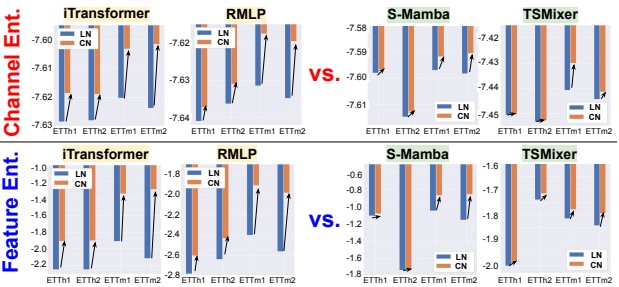

Figure 5: **Entropy gain by CN.** Non-CID models show greater channel/feature entropy gains from CN than CID models.

## 6.2. Entropy Analysis

We demonstrate the effect of our method through entropy analyses with four different backbones, showing that it 1) enriches feature representations (*feature entropy* ↑), 2) increases the uniqueness of each channel representation (*channel entropy* ↑), and 3) diversifies the attention heads and correlation between channel representations.

**Gaussian entropy.** Let $Z \in \mathbb{R}^D$ be a random vector following a multivariate Gaussian distribution with a covariance matrix $\Sigma \in \mathbb{R}^{D \times D}$. Then, the Gaussian entropy of $Z$ is defined as $H(Z) = \frac{1}{2} \log\left((2\pi e)^D \det(\Sigma)\right)$, which can be estimated by $N$ samples $\mathbf{z} = [\mathbf{z}_1, \mathbf{z}_2, \ldots, \mathbf{z}_N]^\top \in \mathbb{R}^{N \times D}$ as $H(\mathbf{z}) = \frac{1}{2} \log\left((2\pi e)^D \det\left(\frac{1}{N}\mathbf{z}^\top\mathbf{z} + \varepsilon \boldsymbol{I}\right)\right)$, with $\varepsilon \boldsymbol{I}$ added to avoid non-trivial solutions, following the previous works (Yu et al., 2020; Chen et al., 2024b; 2025).

For the analysis, we compute the average over a test dataset with $\bar{\mathbf{z}} \in \mathbb{R}^{C \times D}$ and use the normalized entropy averaged over the last dimension for comparison across different dimensions. Then, we define the entropy of $\bar{\mathbf{z}}$ and $\bar{\mathbf{z}}^\top$ as the *feature entropy* and *channel entropy* respectively, as they measure 1) the richness of the feature dimension and 2) uniqueness of each channel representation.

**Entropy gain of non-CID vs. CID models.** Figure 5 illustrates the gains in channel and feature entropies achieved by

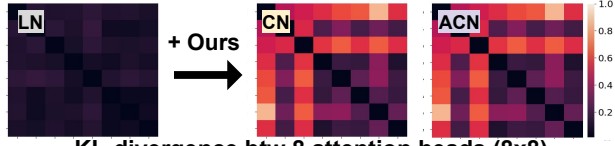

Figure 6: Diversity of attention heads.

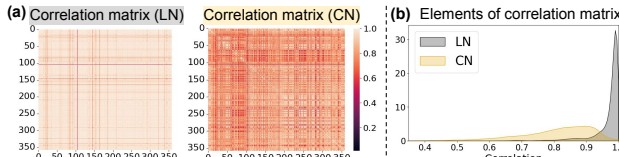

Figure 7: Diversity of correlations btw channel representations.

CN for both non-CID and CID models. The figure shows that non-CID models exhibit higher gains compared to CID models, indicating *richer* feature representations and *greater uniqueness* in channel representations. This supports our argument that the proposed method benefits non-CID models more than CID models, which aligns with the greater performance gain of non-CID models, as shown in Figure 2(a).

**Entropy gain by datasets.** To evaluate the effectiveness of CN across datasets, we analyze the channel entropy gain achieved by CN using iTransformer with respect to (1) the number of channels and (2) the performance gain for each dataset. Figure 4(a) illustrates the relationship between the entropy gain and $C$, showing that datasets with higher $C$ achieve greater entropy gain. Figure 4(b) presents the relationship between the entropy gain and MSE improvement, with a correlation ($\rho$) of 0.724, indicating that datasets with higher entropy gain show greater performance improvement.

**Diverse attention heads & correlations btw channels.** Figure 6 illustrates the KL divergence (KLD) between the distributions of eight attention heads of iTransformer on PEMS03 (Chen et al., 2001), showing that our method enables the model to maintain greater diversity across the heads. Specifically, the average KLD between the heads of the first and last encoder layers is 0.289 and 0.077 for

**(a) t-SNE of CN parameters**

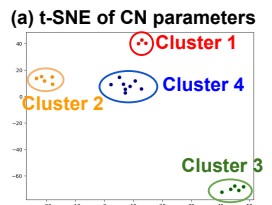

**(b) Visualization of 3 channels for each cluster (Cluster 1, 2, 3, 4)**

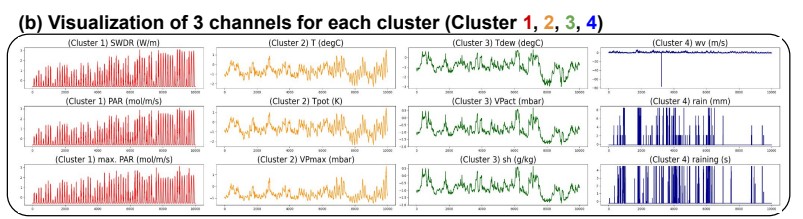

**(c) Correlation matrix of 21 channels**

Figure 8: **Visualization of parameters and channels.** (a) shows the t-SNE of the parameters of CN, with four clusters formed. (b) visualizes three channels from each cluster, demonstrating that channels in the same cluster share similar patterns except for those in the 4th cluster. (c) visualizes the correlation matrix of the channels, where channels in the 4th cluster lack close relationships with others.

| $L = 96, H = 12$ | iTrans. | + Ch. identifier | + C-LoRA | Ours | |
|---|---|---|---|---|---|
| | | | | + CN | + ACN |
| Train (sec/epoch) | 7.7 | 9.4 | 11.1 | 7.8 | 10.8 |
| Inference (ms) | 2.0 | 2.3 | 2.8 | 2.1 | 2.5 |
| # Parameters | 3.2M | + 0.1M | +2.8M | + 0.7M | +1.4M |
| Avg. MSE | .254 | .168 | .169 | .159 | **.153** |

Table 8: Efficiency analysis.

LN, compared to 0.369 and 0.395 for CN. Additionally, Figure 7(a) presents the correlation matrices of channel representations of PEMS03 using iTransformer with and without CN, along with the distribution of matrix elements in Figure 7(b), demonstrating that CN enhances the diversity of the correlations. This increase in diversity in both aspects supports the performance improvements achieved by our method, with the average MSE across four horizons being 0.142, 0.101, and 0.098 for LN, CN, and ACN.

### 6.3. Other Analyses

**Visualization of CN params.** To demonstrate that the parameters of CN effectively capture the CID, we visualize the parameters ($\alpha$) of 21 channels in Weather (Wu et al., 2021) using t-SNE (Van der Maaten & Hinton, 2008). Figure 8 shows the result, displaying (a) four distinct clusters and (b) the visualization of channels corresponding to each cluster. The figure indicates that channels with similar patterns belong to the same cluster, except for the fourth cluster (blue), whose channels show no close relationship with other channels, as also shown by the (c) correlation matrix.

**Efficiency analysis.** Table 8 shows the 1) number of parameters, 2) training time (per epoch), and 3) inference time (per data instance) of iTransformer on PEMS08 (Chen et al., 2001) across various methods for CID. The results indicate that applying our methods has minimal impact on the number of parameters and computational time, while providing a greater performance gain compared to other methods.

**Performance under varying $L$s.** To validate the effectiveness of our method under various sizes of lookback windows ($L$), we evaluate our method on iTransformer with a forecast horizon of $H = 12$ for the PEMS datasets and $H = 96$ for the other datasets. Figure 9 indicates that the performance gain remains robust across all datasets regardless of $L$.

**Various $K$s for PCN.** Figure 10 shows the t-SNE visualizations of prototype parameters ($\alpha^P$) of PCN across varying

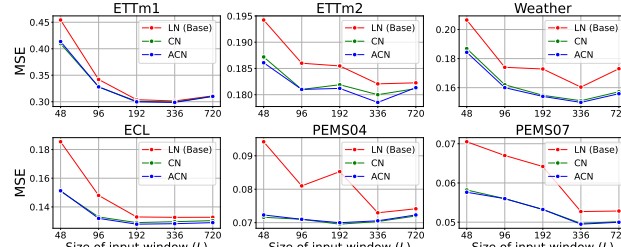

Figure 9: Effectiveness of CN/ACN under various $L$.

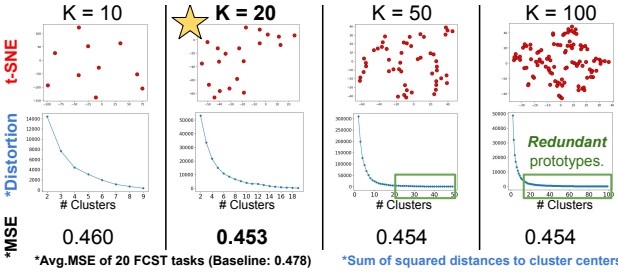

Figure 10: t-SNE & distortion plot of PCN parameters.

numbers of prototypes ($K$), using UniTS as the backbone. The distortion plots, shown below, are obtained by performing K-means clustering on these parameters to assess the redundancy of the prototypes. The figures indicate that increasing $K$ leads to performance stabilization after a certain point ($K = 20$), as redundant prototypes begin to emerge.

For further analyses, please refer to the below sections:

- Theoretical entropy analysis: Appendix C
- Comparison with Instance Normalization: Appendix F
- Robustness to $K$, $\tau$, similarity space: Appendix I, J, K
- PCN for zero-shot forecasting with TSFM: Appendix L
- Application of multiple methods for CID: Appendix E
- Visualization of TS forecasting results: Appendix O

## 7. Conclusion

In this work, we introduce CN, a normalization strategy to enhance CID of TS models with channel-specific parameters. Furthermore, we propose ACN to adapt to input TS on single-task models, and PCN to handle multiple datasets with unknown/varying number of channels on TSFMs. A potential direction for future work involves developing a method to automatically determine the number of prototypes for PCN based on the dataset. We hope that our work highlights the importance of CID in TS analysis.

## Impact Statement

This paper aims to advance time series modeling by emphasizing the importance of channel identification and proposing Channel Normalization to enhance it. We do not identify any specific societal consequences that require emphasis.

## Acknowledgements

This work was supported by the National Research Foundation of Korea (NRF) grant funded by the Korea government (MSIT) (2020R1A2C1A01005949, 2022R1A4A1033384, RS-2023-00217705, RS-2024-00341749), the MSIT(Ministry of Science and ICT), Korea, under the ICAN(ICT Challenge and Advanced Network of HRD) support program (RS-2023-00259934) supervised by the IITP(Institute for Information & Communications Technology Planning & Evaluation), Yonsei University Research Fund (2024-22-0148), and a grant from KRAFTON AI.

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

# Appendix

# A. Experimental Settings

## A.1. Dataset Statistics

**Dataset statistics.** For the experiments, 12 datasets from various domains are used, with their statistics detailed in Table A.1, where $C$ and $T$ represent the number of channels and timesteps, respectively.

**Dataset split.** We follow the same data processing steps and train-validation-test split protocol as used in S-Mamba (Wang et al., 2025), maintaining a chronological order in the separation of training, validation, and test sets, using a 6:2:2 ratio for the Solar-Energy, ETT, and PEMS datasets, and a 7:1:2 ratio for the other datasets. Hyperparameters are tuned based on the validation loss.

**Size of lookback window ($L$).** Following the previous works (Nie et al., 2024; Liu et al., 2024), $L$ is uniformly set to 96 for all datasets and models. Further analysis regarding the performance under different $L$ is discussed in Figure 9.

| Dataset | Statistics | | Dataset Split | Size of Input & Output | |
| --- | --- | --- | --- | --- | --- |
| | $C$ | $T$ | $(N_{\text{train}}, N_{\text{val}}, N_{\text{test}})$ | $L$ | $H$ |
| ETTh1 (Zhou et al., 2021) | | 17420 | (8545, 2881, 2881) | | |
| ETTh2 (Zhou et al., 2021) | 7 | 17420 | (8545, 2881, 2881) | | |
| ETTm1 (Zhou et al., 2021) | | 69680 | (34465, 11521, 11521) | | |
| ETTm2 (Zhou et al., 2021) | | 69680 | (34465, 11521, 11521) | | |
| Exchange (Wu et al., 2021) | 8 | 7588 | (5120, 665, 1422) | | {96, 192, 336, 720} |
| Weather (Wu et al., 2021) | 21 | 52696 | (36792, 5271, 10540) | | |
| ECL (Wu et al., 2021) | 321 | 26304 | (18317, 2633, 5261) | 96 | |
| Solar-Energy (Lai et al., 2018) | 137 | 52560 | (36601, 5161, 10417) | | |
| PEMS03 (Liu et al., 2022) | 358 | 26209 | (15617, 5135, 5135) | | |
| PEMS04 (Liu et al., 2022) | 307 | 15992 | (10172, 3375, 3375) | | {12, 24, 48, 96} |
| PEMS07 (Liu et al., 2022) | 883 | 28224 | (16911, 5622, 5622) | | |
| PEMS08 (Liu et al., 2022) | 170 | 17856 | (10690, 3548, 3548) | | |

Table A.1: Datasets for TS forecasting.

## A.2. Experimental Setups

**Application of CN/ACN.** For all experiments regarding TS forecasting with four different backbones, we use the official code from C-LoRA (Nie et al., 2024), except for S-Mamba (Wang et al., 2025), as C-LoRA does not use S-Mamba as a backbone.

**Application of PCN.** For all experiments involving TSFM, UniTS (Gao et al., 2024) is trained across multiple tasks using a unified protocol. To accommodate the largest dataset, samples from each dataset are repeated within each epoch. The training protocol, as outlined in the original paper, is as follows:

- **Supervised training:** Models are trained for 5 epochs with gradient accumulation, yielding an effective batch size of 1024. The initial learning rate is set to 3.2e-2 and adjusted using a multi-step decay schedule.
- **Self-supervised pretraining:** Models are trained for 10 epochs with an effective batch size of 4096, starting with a learning rate of 6.4e-3 and utilizing a cosine decay schedule.

The embedding dimension is set to 64 for the supervised version and 32 for the prompt-tuning version. Note that we encountered a convergence issue in the prompt-tuning setting, which was also reported by others in a GitHub issue. To resolve this, we set the hidden dimension to 32, which led to a performance decrease compared to the results in the original paper. For a fair comparison, this setting is applied uniformly to both UniTS and its application to PCN.

**Parameter initialization.** The initialization of the parameters for Channel Normalization (CN), Adaptive Channel Normalization (ACN), and Prototypical Channel Normalization (PCN) is designed to ensure that no normalization occurs when learning has not yet taken place:

- The **scale** parameter ($\alpha$) is initialized to 1.
- The **shift** parameter ($\beta$) is initialized to 0.

This choice is consistent with the default initialization used in PyTorch (Paszke et al., 2019) normalization layers, including Layer Normalization and Batch Normalization. Therefore, the parameters for CN, ACN, and PCN are initialized as follows:

- CN: $\alpha = 1, \beta = 0$
- ACN: $\alpha_G = 1, \alpha_L = 0, \beta_G = 1, \beta_L = 0$
- PCN: $\alpha_P = 1, \beta_P = 0$

# B. Properties of TS Backbones

## B.1. Channel Identifiability

**Definition.** A MTS forecasting model $f : \mathbb{R}^{L \times C} \to \mathbb{R}^{H \times C}$ exhibits *channel identifiability* (CID) if, for any input TS $\mathbf{x} \in \mathbb{R}^{L \times C}$ the output $f(\mathbf{x})$ depends on the channel index $c$ of $\mathbf{x}$, such that the forecasted value $f(\mathbf{x})_{:,c}$ is unique to $c$, even when all channels in $\mathbf{x}$ have identical values.

Using the above property, MTS forecasting models can be classified into two categories: models without CID and models with CID.

**1) Model without channel identifiability.** If a model $f$ lacks CID, then for any $\mathbf{x} \in \mathbb{R}^{L \times C}$ with all channels having identical input values, the forecasted outputs for all channels will also be identical:

$$\mathbf{x}\left[:, c_1\right] = \mathbf{x}\left[:, c_2\right] \Rightarrow f(\mathbf{x})\left[:, c_1\right] = f(\mathbf{x})\left[:, c_2\right], \quad \forall c_1, c_2. \tag{B.1}$$

**2) Model with channel identifiability.** If a model $f$ possesses CID, then for any $\mathbf{x} \in \mathbb{R}^{L \times C}$ with all channels having identical input values, the forecasted outputs for different channels will be distinct due to the model's ability to incorporate channel positional information:

$$\mathbf{x}\left[:, c_1\right] = \mathbf{x}\left[:, c_2\right] \nRightarrow f(\mathbf{x})\left[:, c_1\right] = f(\mathbf{x})\left[:, c_2\right], \quad \forall c_1, c_2, \tag{B.2}$$

where the inequality arises from the model's recognition of channel positions.

## B.2. Data Dependency

**Definition.** A MTS forecasting model $f : \mathbb{R}^{L \times C} \to \mathbb{R}^{T' \times C}$ exhibits data dependency if the model parameters $\theta$ depend on the input TS $\mathbf{x}$. Specifically, for a given input TS $\mathbf{x} \in \mathbb{R}^{L \times C}$, the model parameters $\theta$ may vary based on the content or structure of $\mathbf{x}$, affecting the model's output.

Using the above property, MTS forecasting models can be classified into two categories: models without data dependency and models with data dependency.

**1) Model without data dependency.** If a model $f$ does not exhibit data dependency, then the model parameters $\theta$ are fixed and independent of the input TS $\mathbf{x}$:

$$\mathbf{y} = f\left(\mathbf{x}, \theta\right). \tag{B.3}$$

**2) Model with data dependency.** If a model $f$ exhibits data dependency, then the model parameters $\theta$ depend on the input TS $\mathbf{x}$:

$$\mathbf{y} = f\left(\mathbf{x}, \theta(\mathbf{x})\right). \tag{B.4}$$

## C. Theoretical Entropy Analysis

Following the previous work (Chen et al., 2025), we analyze our approach using theoretical entropy analysis.

**Justification 1.** Applying CN achieves a more informative representation ($\mathbf{Z}_{\text{CN}}$) compared to LN ($\mathbf{Z}_{\text{LN}}$) or without any normalization ($\mathbf{Z}_{\text{None}}$), as it increases in the entropy :

$$H(\mathbf{Z}_{\text{None}}) \leq H(\mathbf{Z}_{\text{LN}}) \leq H(\mathbf{Z}_{\text{CN}}). \tag{C.1}$$

*Proof.* The joint entropy can be decomposed as follows:

$$\underbrace{H(\mathbf{Z})}_{=H_{\text{None}}} \leq H(\mathbf{Z}) + H(\alpha_1, \beta_1 | \mathbf{Z}) \tag{C.2}$$

$$= \underbrace{H(\mathbf{Z}, \alpha_1, \beta_1)}_{=H_{\text{LN}}} \tag{C.3}$$

$$\leq H(\mathbf{Z}, \alpha_1, \beta_1) + H(\{\alpha_i, \beta_i\}_{i=2}^{C} | \alpha_1, \beta_1) \tag{C.4}$$

$$= \underbrace{H(\mathbf{Z}, \{\alpha_i, \beta_i\}_{i=1}^{C})}_{=H_{\text{CN}}}, \tag{C.5}$$

This follows from the non-negativity of conditional entropy (Thomas & Joy, 2006).

**Justification 2.** A more informative representation (i.e., higher $H(\mathbf{Z})$) can potentially lower forecasting error, as under the Gaussian assumption, the minimum mean-squared error (MMSE) is bounded by:

$$\text{MMSE} \geq \frac{\exp\left(2H\left(\mathbf{Y} \mid \mathbf{Z}\right)\right)}{2\pi e}. \tag{C.6}$$

*Proof.* Following Equation 1, we construct the chain with a modification where the last layer of $g_2$ is separated:

$$\mathbf{X} \xrightarrow{g_1} \mathbf{Z}_{\text{pre}} \xrightarrow{f} \mathbf{Z} \xrightarrow{g_2'} \mathbf{Z}_{\text{post}} \xrightarrow{g_2''} \hat{\mathbf{Y}}. \tag{C.7}$$

This allows the propagation in the final layer $g_2$ to be expressed as:

$$\hat{\mathbf{Y}} = g_2''(\mathbf{Z}_{\text{post}}) = \mathbf{W}\mathbf{Z}_{\text{post}}. \tag{C.8}$$

Assuming a Gaussian distribution for $\mathbf{Z}_{\text{post}}$, $\hat{\mathbf{Y}}$, and $\mathbf{Y}$, we can derive the following bound, as shown in previous works (Carson et al., 2012; Prasad, 2010):

$$\text{MMSE} \geq \frac{\exp 2H\left(\mathbf{Y} \mid \mathbf{Z}_{\text{post}}\right)}{2\pi e}. \tag{C.9}$$

Since $\mathbf{Z}_{\text{post}} = g_2'(\mathbf{Z})$, the chain property (Thomas & Joy, 2006) ensures that $\mathbf{Z}$ contains at least as much information about $\mathbf{Y}$ as $\mathbf{Z}_{\text{post}}$, i.e., knowing $\mathbf{Z}$ reduces the uncertainty about $\mathbf{Y}$:

$$H\left(\mathbf{Y} \mid \mathbf{Z}_{\text{post}}\right) \geq H\left(\mathbf{Y} \mid \mathbf{Z}\right). \tag{C.10}$$

By substituting Equation C.10 into Equation C.9, we obtain:

$$\text{MMSE} \geq \frac{\exp\left(2H\left(\mathbf{Y} \mid \mathbf{Z}_{\text{post}}\right)\right)}{2\pi e} \geq \frac{\exp\left(2H\left(\mathbf{Y} \mid \mathbf{Z}\right)\right)}{2\pi e}. \tag{C.11}$$

# D. Various Backbones

**Backbones for CN/ACN.** The four backbones used in the experiments are categorized based on their 1) channel identifiability (CID) and 2) data dependency, as shown in Figure D.1.

- RMLP (Li et al., 2023) captures temporal dependencies within each channel in a channel-independent manner, applying identical weights across all channels.
- iTransformer (Liu et al., 2024) captures channel dependencies using a self-attention mechanism that is order-invariant, rendering channels unidentifiable (non-CID).
- TSMixer (Chen et al., 2023) employs MLPs to capture both temporal and channel dependencies, with distinct weights assigned to each channel.
- S-Mamba (Wang et al., 2025) utilizes the Mamba architecture to capture channel dependencies, leveraging the inherent properties of Mamba (e.g., state-space modeling) to differentiate between channels.

For architectures that utilize Layer Normalization (LN), such as iTransformer and S-Mamba, we replace LN with our proposed method. For architectures without any normalization, we incorporate our method directly.

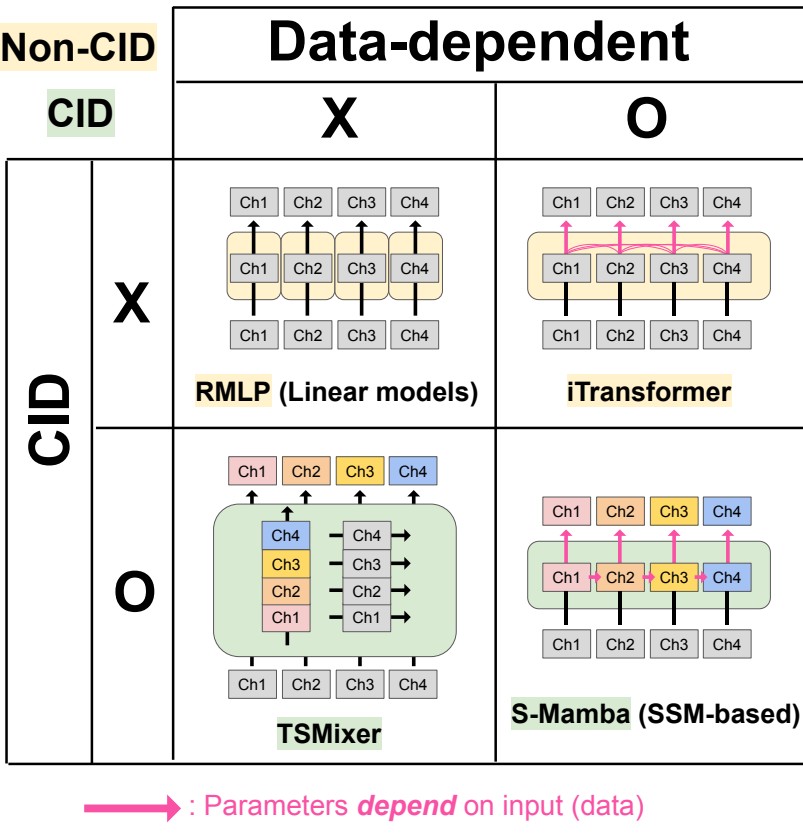

Figure D.1: Four different backbones and their properties.

**Backbones for PCN.** UniTS (Gao et al., 2024) is designed with three distinct UniTS blocks, as well as a `GEN` tower and a `CLS` tower. Each data source is assigned unique prompt and task tokens, while tasks within the same source that require different forecast lengths use a shared prompt and `GEN` token. To facilitate zero-shot learning for new datasets, a universal prompt and `GEN` token are utilized across all data sources.

## E. Application of Multiple Methods for CID

As a plug-in method, ACN is applicable to TS models along with other CID methods. To demonstrate that our method complements these techniques, we evaluate its performance when combined with *channel identifier* (Chi et al., 2024) and *C-LoRA* (Nie et al., 2024), using iTransformer (Liu et al., 2024) as the backbone, as shown in Table E.1.

| iTransformer | Average MSE across 4 horizons | | | | | | | | | | | | Avg. |
|---|---|---|---|---|---|---|---|---|---|---|---|---|---|
| | ETTh1 | ETTh2 | ETTm1 | ETTm2 | PEMS03 | PEMS04 | PEMS07 | PEMS08 | Exchange | Weather | Solar | ECL | |
| - | .457 | .384 | .408 | .293 | .142 | .121 | .102 | .254 | .368 | .260 | .234 | .179 | .275 |
| + ACN | **.438** | **.374** | .395 | .288 | **.098** | **.088** | .085 | **.153** | .349 | **.245** | **.220** | **.158** | **.241** |
| + ACN + C-LoRA | **.438** | .380 | .397 | **.285** | .109 | .090 | .096 | .162 | .360 | **.245** | .227 | .162 | .246 |
| + ACN + Channel identifier | .442 | .377 | **.394** | .289 | .099 | .089 | **.084** | **.158** | **.344** | **.245** | .222 | **.158** | .242 |
| + ACN + C-LoRA + Channel identifier | .440 | .381 | .400 | .286 | .105 | .090 | .089 | .163 | .346 | **.245** | .226 | .162 | .244 |

Table E.1: Application of multiple methods for CID.

## F. Comparison with Instance Normalization

Table F.1 shows the comparison of our methods (CN, ACN) with Instance Normalization (IN) (Ulyanov, 2016) on iTransformer (Liu et al., 2024) in terms of average MSE across four horizons for various datasets, demonstrating that our methods outperforms IN.

| iTransformer | Average MSE across 4 horizons | | | | | | | | | | | | Avg. |
|---|---|---|---|---|---|---|---|---|---|---|---|---|---|
| | ETTh1 | ETTh2 | ETTm1 | ETTm2 | PEMS03 | PEMS04 | PEMS07 | PEMS08 | Exchange | Weather | Solar | ECL | |
| + IN | .442 | .377 | .397 | .291 | .101 | .092 | .088 | .165 | .356 | .249 | .226 | .162 | .246 |
| + CN | .441 | .376 | .396 | .289 | .101 | **.088** | .087 | .159 | .352 | .247 | .228 | .161 | .244 |
| + ACN | **.438** | **.374** | **.395** | **.288** | **.098** | **.088** | **.085** | **.153** | **.349** | **.245** | **.220** | **.158** | **.241** |

Table F.1: Comparison with Instance Normalization (IN).

## G. Comparison with Constant Vectors

Table G.1 presents the performance of adding different constant vectors to each channel token, allowing the model to distinguish channels on iTransformer (Liu et al., 2024). The results indicate that this simple addition improves performance, while our methods (CN, ACN) achieves better performance in terms of average MSE across four horizons for various datasets.

| iTransformer | Average MSE across 4 horizons | | | | | | | | | | | | Avg. | Imp. |
|---|---|---|---|---|---|---|---|---|---|---|---|---|---|---|
| | ETTh1 | ETTh2 | ETTm1 | ETTm2 | PEMS03 | PEMS04 | PEMS07 | PEMS08 | Exchange | Weather | Solar | ECL | | |
| - | .457 | .384 | .408 | .293 | .142 | .121 | .102 | .254 | .368 | .260 | .234 | .179 | .275 | - |
| + Constant vector | .443 | .378 | .397 | .290 | .114 | .113 | .103 | .181 | .355 | .246 | .233 | .170 | .252 | 8.4% |
| + CN | .441 | .376 | .396 | .289 | .101 | .088 | .087 | .159 | .352 | .247 | .228 | .161 | .244 | 11.3% |
| + ACN | **.438** | **.374** | **.395** | **.288** | **.098** | **.088** | **.085** | **.153** | **.349** | **.245** | **.220** | **.158** | **.241** | 12.4% |

Table G.1: Comparison with contant vectors.

## H. Robustness to Similarity Metric

To construct a channel similarity matrix $S \in \mathbb{R}^{B \times C \times C}$ for ACN, various similarity metric can be employed. To evaluate whether the proposed method is sensitive to the choice of similarity metric, we compare several options, including (negative) cosine similarity, $\ell_1$ distance, and $\ell_2$ distance. Table H.1 presents the average MSE across four horizons for various datasets, demonstrating that the performance remains robust to the choice of similarity metric.

| iTransformer | | Average MSE across 4 horizons | | | | | | | | Avg. |
|---|---|---|---|---|---|---|---|---|---|---|
| | | ETTh1 | ETTh2 | ETTm1 | ETTm2 | Exchange | Weather | Solar | ECL | |
| LN (Base) | | .457 | .384 | .408 | .293 | .368 | .260 | .234 | .179 | .329 |
| CN | $\ell_1$ | .440 | **.374** | **.395** | **.288** | .350 | **.245** | **.220** | **.179** | .309 |
| | $\ell_2$ | .439 | .375 | **.395** | **.288** | .350 | **.245** | .221 | **.158** | .309 |
| | Cosine | **.438** | **.374** | **.395** | **.288** | **.349** | **.245** | **.220** | **.158** | .308 |

Table H.1: Robustness to similarity metric for ACN.

## I. Robustness to Number of Prototypes $K$

Table I.1 shows the results of applying PCN with various values of $K$. The results indicate that the performance remains robust to the choice of $K$.

| PCN | K | Average MSE across 4 horizons | | | | | | | | | | | | Avg. |
|---|---|---|---|---|---|---|---|---|---|---|---|---|---|---|
| | | ETTh1 | ETTh2 | ETTm1 | ETTm2 | PEMS03 | PEMS04 | PEMS07 | PEMS08 | Exchange | Weather | Solar | ECL | |
| ✗ | 1 | .457 | .384 | .408 | .293 | .142 | .121 | .102 | .254 | .368 | .260 | .234 | .179 | .275 |
| | 2 | .440 | .376 | .404 | .290 | **.115** | .121 | .102 | .180 | .340 | **.257** | .235 | **.169** | **.252** |
| | 3 | .439 | **.375** | **.403** | .290 | .116 | .121 | .101 | .179 | .345 | **.257** | .235 | **.169** | **.252** |
| ✓ | 5 | .437 | .376 | .404 | .289 | .117 | .120 | **.101** | **.176** | .349 | **.257** | .232 | **.169** | **.252** |
| | 10 | .438 | .376 | **.403** | .289 | .116 | **.119** | **.101** | .182 | .339 | **.257** | .233 | **.169** | **.252** |
| | 20 | **.434** | .376 | .404 | **.288** | .117 | .120 | .102 | .183 | **.336** | **.257** | .233 | **.169** | **.252** |

Table I.1: Robustness to $K$ for PCN.

## J. Robustness to Temperature $\tau$

Tables J.1, J.2, J.3, and J.4 display the average MSE across four different horizons for the 12 datasets, with four different backbones, using various values of the temperature ($\tau$) in ACN. The results show that the effectiveness of ACN is consistent across different values of $\tau$.

| $\tau$ | Average MSE across 4 horizons | | | | | | | | | | | | Avg. |
| --- | --- | --- | --- | --- | --- | --- | --- | --- | --- | --- | --- | --- | --- |
| | ETTh1 | ETTh2 | ETTm1 | ETTm2 | PEMS03 | PEMS04 | PEMS07 | PEMS08 | Exchange | Weather | Solar | ECL | |
| 0.05 | **.439** | .376 | .396 | **.288** | .099 | **.088** | .087 | .156 | .351 | **.245** | **.221** | .159 | **.242** |
| 0.1 | **.439** | .376 | .396 | **.288** | .099 | **.088** | .085 | .156 | .351 | .246 | .222 | **.158** | **.242** |
| 0.2 | **.439** | .375 | .395 | .289 | .099 | **.088** | .086 | **.154** | .350 | .246 | .223 | **.158** | **.242** |
| 0.5 | **.439** | .375 | .395 | .289 | **.098** | **.088** | .086 | .159 | **.350** | .247 | .224 | **.158** | **.242** |
| 1.0 | **.439** | .376 | .395 | .289 | **.098** | **.088** | .087 | .159 | **.350** | .248 | .224 | **.158** | **.242** |

Table J.1: Robustness to $\tau$ for ACN with **iTransformer**.

| | Backbone: S-Mamba | | | | | | | | | | | | |
| --- | --- | --- | --- | --- | --- | --- | --- | --- | --- | --- | --- | --- | --- |
| $\tau$ | Average MSE across 4 horizons | | | | | | | | | | | | Avg. |
| | ETTh1 | ETTh2 | ETTm1 | ETTm2 | PEMS03 | PEMS04 | PEMS07 | PEMS08 | Exchange | Weather | Solar | ECL | |
| 0.05 | **.449** | **.375** | **.394** | **.285** | .108 | .093 | .075 | **.122** | .358 | **.248** | **.228** | .164 | **.241** |
| 0.1 | **.449** | **.375** | **.394** | **.285** | .107 | .095 | **.074** | .127 | .358 | **.248** | .229 | .168 | **.241** |
| 0.2 | **.449** | **.375** | .395 | **.285** | .108 | .095 | .076 | .129 | .358 | .249 | .230 | **.163** | **.241** |
| 0.5 | **.449** | **.375** | .395 | **.285** | .109 | .095 | .076 | .124 | .358 | .250 | .231 | .165 | **.241** |
| 1.0 | **.449** | **.375** | .395 | **.285** | .108 | .095 | .076 | .124 | **.358** | .250 | .231 | .165 | **.241** |

Table J.2: Robustness to $\tau$ for ACN with **S-Mamba**.

| | Backbone: TSMixer | | | | | | | | | | | | |
| --- | --- | --- | --- | --- | --- | --- | --- | --- | --- | --- | --- | --- | --- |
| $\tau$ | Average MSE across 4 horizons | | | | | | | | | | | | Avg. |
| | ETTh1 | ETTh2 | ETTm1 | ETTm2 | PEMS03 | PEMS04 | PEMS07 | PEMS08 | Exchange | Weather | Solar | ECL | |
| 0.05 | **.453** | **.387** | .391 | **.280** | .130 | .112 | .105 | .178 | **.356** | **.242** | .248 | **.174** | .255 |
| 0.1 | **.453** | **.387** | .391 | **.280** | .124 | **.109** | **.103** | .177 | **.356** | **.242** | .247 | **.174** | .254 |
| 0.2 | **.453** | .388 | .391 | **.280** | **.120** | **.109** | **.103** | **.168** | **.356** | **.242** | .246 | .175 | .253 |
| 0.5 | .455 | .388 | **.390** | **.280** | .121 | .110 | **.103** | .171 | **.356** | **.242** | **.246** | **.174** | .253 |
| 1.0 | .455 | .388 | **.390** | **.280** | .122 | .110 | .104 | .173 | **.356** | **.242** | **.246** | **.174** | .254 |

Table J.3: Robustness to $\tau$ for ACN with **TSMixer**.

| | Backbone: RMLP | | | | | | | | | | | | |
| --- | --- | --- | --- | --- | --- | --- | --- | --- | --- | --- | --- | --- | --- |
| $\tau$ | Average MSE across 4 horizons | | | | | | | | | | | | Avg. |
| | ETTh1 | ETTh2 | ETTm1 | ETTm2 | PEMS03 | PEMS04 | PEMS07 | PEMS08 | Exchange | Weather | Solar | ECL | |
| 0.05 | .449 | .378 | .384 | **.277** | .162 | .164 | .136 | .200 | .354 | **.246** | .244 | **.189** | .265 |
| 0.1 | .450 | **.377** | .384 | **.277** | **.159** | **.157** | **.131** | **.187** | .354 | **.246** | .243 | **.189** | **.263** |
| 0.2 | .450 | **.377** | .383 | **.277** | .160 | **.157** | **.131** | .188 | **.353** | .247 | .251 | **.189** | .264 |
| 0.5 | **.448** | **.377** | .384 | **.277** | .178 | .168 | .136 | .199 | **.353** | .247 | .257 | .190 | .268 |
| 1.0 | **.448** | **.377** | .384 | **.277** | .180 | .168 | .138 | .202 | **.353** | .247 | .257 | .191 | .268 |

Table J.4: Robustness to $\tau$ for ACN with **RMLP**.

# K. Robustness to Similarity Space for ACN & PCN

## K.1. Similarity Space for ACN

The similarity between the channels in TS for ACN can be calculated either in the data space ($X$) or the latent space ($Z$). Table K.1 indicates that performance is robust to the choice of space across various datasets with four different backbones, further validating the effectiveness of ACN.

| Sim. space | | | ETTh1 | ETTh2 | ETTm1 | ETTm2 | PEMS03 | PEMS04 | PEMS07 | PEMS08 | Exchange | Weather | Solar | ECL | Avg. |
|---|---|---|---|---|---|---|---|---|---|---|---|---|---|---|---|
| iTrans. | | - | .457 | .384 | .408 | .293 | .142 | .121 | .102 | .254 | .368 | .260 | .234 | .179 | .275 |
| | ACN | X | **.438** | .375 | **.395** | **.288** | .100 | .089 | **.085** | .156 | **.349** | .247 | .221 | **.158** | .242 |
| | | Z | **.438** | **.374** | **.395** | **.288** | **.098** | **.088** | **.085** | **.153** | **.349** | **.245** | **.220** | **.158** | **.241** |
| S-Mam. | | - | .457 | .383 | .398 | .290 | .133 | .096 | .090 | .157 | .364 | .252 | .244 | .174 | .253 |
| | ACN | X | **.448** | .375 | **.394** | .285 | **.107** | **.092** | **.073** | **.121** | **.357** | **.247** | **.228** | **.162** | **.240** |
| | | Z | **.448** | **.374** | **.394** | .284 | **.107** | **.092** | **.073** | **.121** | **.357** | **.247** | **.228** | **.162** | **.240** |
| TSMixer | | - | .462 | .403 | .401 | .287 | .129 | .115 | .115 | .186 | .365 | .260 | .255 | .211 | .266 |
| | ACN | X | **.453** | .387 | .387 | **.280** | .121 | **.109** | **.103** | .168 | **.356** | **.242** | **.245** | .178 | .244 |
| | | Z | **.453** | **.386** | **.385** | **.280** | .120 | **.109** | **.103** | **.167** | **.356** | **.242** | **.245** | .174 | **.243** |
| RMLP | | - | .471 | .381 | .401 | .280 | .205 | .236 | .200 | .277 | .356 | .272 | .261 | .228 | .297 |
| | ACN | X | **.448** | **.376** | **.383** | **.277** | .161 | .157 | .131 | **.186** | **.353** | **.246** | .243 | **.189** | **.262** |
| | | Z | **.448** | **.376** | **.383** | **.277** | .159 | .156 | .131 | .187 | **.353** | **.246** | .242 | **.189** | **.262** |

Table K.1: Robustness to similarity space for ACN.

### K.2. Similarity Space for PCN

The similarity between the channels in TS and the prototypes for PCN can be calculated either in the data space ($X$), latent space ($Z$), or the latent space with an additional linear layer ($h$), which is used to align the space between the input TS and the prototypes. Table K.2 indicates that performance is robust to the choice of space across various datasets with different numbers of prototypes ($K$), further validating the effectiveness of PCN.

| PCN ($K = 5$) | | Average MSE across 4 horizons | | | | | | Avg. |
|---|---|---|---|---|---|---|---|---|
| | | ETTh1 | ETTh2 | ETTm1 | ETTm2 | Exchange | Weather | |
| | - | .457 | .384 | .408 | .293 | .368 | .260 | .362 |
| Space | $X$ | .440 | .378 | **.404** | **.289** | .342 | .258 | **.352** |
| | $Z$ | .443 | .381 | **.404** | .290 | **.340** | .259 | .353 |
| | $h(Z)$ | **.437** | **.376** | **.404** | **.289** | .349 | **.257** | **.352** |

| PCN ($K = 10$) | | Average MSE across 4 horizons | | | | | | Avg. |
|---|---|---|---|---|---|---|---|---|
| | | ETTh1 | ETTh2 | ETTm1 | ETTm2 | Exchange | Weather | |
| | - | .457 | .384 | .408 | .293 | .368 | .260 | .362 |
| Space | $X$ | .440 | **.377** | .406 | **.289** | .342 | .259 | .352 |
| | $Z$ | .443 | .378 | .405 | .290 | .341 | .260 | .355 |
| | $h(Z)$ | **.438** | **.377** | **.403** | **.289** | **.339** | **.257** | **.351** |

| PCN ($K = 20$) | | Average MSE across 4 horizons | | | | | | Avg. |
|---|---|---|---|---|---|---|---|---|
| | | ETTh1 | ETTh2 | ETTm1 | ETTm2 | Exchange | Weather | |
| | - | .457 | .384 | .408 | .293 | .368 | .260 | .362 |
| Space | $X$ | .439 | .377 | **.404** | .289 | **.333** | .258 | .350 |
| | $Z$ | .441 | .379 | **.404** | .290 | .341 | .259 | .352 |
| | $h(Z)$ | **.434** | **.375** | **.404** | **.288** | .336 | **.257** | **.349** |

Table K.2: Robustness to similarity space for PCN with $K = 5, 10, 20$

## L. Application of PCN to Zero-shot Forecasting with TSFMs

We conduct TS forecasting tasks under two types of zero-shot settings with UniTS (Gao et al., 2024): the *1) Zero-shot dataset*, which involves evaluation on a dataset not seen during training, and the *2) Zero-shot task*, where we evaluate a new forecasting horizon not included in the training process by appending mask tokens at the end of the TS to predict future time steps.

**Zero-shot dataset.** In the TS forecasting task on unseen datasets, we evaluate our method with three datasets (NREL, 2006; McLeod & Gweon, 2013; Hyndman et al., 2008). The results, shown in Table L.1, highlight consistent improvements by incorporating PCN.

**Zero-shot horizon.** For the TS forecasting task with new horizons, we predict an additional 384 time steps beyond the base forecasting horizon of 96 by appending 24 masked tokens of length 16 at the end of the TS. Table L.2 presents the results on four datasets (Zhou et al., 2021; Wu et al., 2021), showing performance improvements across all datasets.

| Dataset | UniTS | | + PCN | | Imp. | |
|---|---|---|---|---|---|---|
| | MSE | MAE | MSE | MAE | MSE | MAE |
| Solar | .597 | .607 | **.592** | **.514** | **0.8%** | **15.3%** |
| River | 1.374 | .698 | **1.272** | **.580** | **7.4%** | **16.9%** |
| Hospital | 1.067 | .797 | **1.046** | **.787** | **2.0%** | **1.3%** |
| Avg. | 1.013 | .701 | **.970** | **.627** | **4.2%** | **10.6%** |

Table L.1: Results of TS forecasting with **zero-shot dataset.**

| Dataset | UniTS | | + PCN | | Imp. | |
|---|---|---|---|---|---|---|
| | MSE | MAE | MSE | MAE | MSE | MAE |
| ECL | .237 | .329 | **.229** | **.322** | **3.4%** | **2.2%** |
| ETTh1 | .495 | .463 | **.486** | **.459** | **1.8%** | **0.9%** |
| Traffic | .632 | .372 | **.616** | **.362** | **2.5%** | **2.7%** |
| Weather | .335 | .336 | **.334** | **.335** | **0.3%** | **0.3%** |
| Avg. | .425 | .375 | **.416** | **.369** | **2.1%** | **1.6%** |

Table L.2: Results of TS forecasting with **zero-shot horizon.**

# M. Full Results: Application of CN & ACN

Table M.1 and Table M.2 present the results of TS forecasting for non-CID and CID models, respectively. The proposed method shows greater improvement in non-CID models, highlighting its role in enabling channel identifiability.

| Models | | iTransformer | | + CN | | + ACN | | RMLP | | + CN | | + ACN | |
|---|---|---|---|---|---|---|---|---|---|---|---|---|---|
| Metric | | MSE | MAE | MSE | MAE | MSE | MAE | MSE | MAE | MSE | MAE | MSE | MAE |
| ETTh1 | 96 | .387 | .405 | .382 | .401 | .381 | .400 | .405 | .413 | .375 | .394 | .381 | .394 |
| | 192 | .441 | .436 | .432 | .429 | .431 | .429 | .460 | .444 | .433 | .426 | .435 | .424 |
| | 336 | .487 | .458 | .472 | .451 | .471 | .450 | .505 | .466 | .479 | .449 | .482 | .446 |
| | 720 | .509 | .494 | .478 | .474 | .470 | .469 | .514 | .490 | .493 | .478 | .492 | .473 |
| | Avg. | .457 | .449 | .441 | .439 | .438 | .438 | .471 | .453 | .445 | .437 | .448 | .435 |
| ETTh2 | 96 | .301 | .350 | .300 | .351 | .299 | .350 | .298 | .349 | .295 | .346 | .291 | .343 |
| | 192 | .381 | .399 | .375 | .397 | .375 | .396 | .374 | .397 | .369 | .395 | .367 | .392 |
| | 336 | .427 | .434 | .410 | .428 | .409 | .427 | .424 | .435 | .423 | .432 | .418 | .429 |
| | 720 | .430 | .446 | .420 | .441 | .413 | .436 | .433 | .449 | .431 | .446 | .428 | .444 |
| | Avg. | .384 | .407 | .376 | .404 | .374 | .402 | .381 | .408 | .380 | .405 | .376 | .402 |
| ETTm1 | 96 | .342 | .377 | .328 | .364 | .328 | .364 | .337 | .374 | .319 | .358 | .318 | .357 |
| | 192 | .383 | .396 | .373 | .388 | .370 | .387 | .379 | .391 | .364 | .383 | .361 | .381 |
| | 336 | .418 | .418 | .409 | .412 | .407 | .411 | .412 | .412 | .394 | .404 | .393 | .403 |
| | 720 | .487 | .456 | .475 | .448 | .474 | .446 | .478 | .447 | .461 | .442 | .459 | .441 |
| | Avg. | .408 | .412 | .396 | .403 | .395 | .402 | .401 | .406 | .384 | .397 | .383 | .396 |
| ETTm2 | 96 | .186 | .272 | .181 | .264 | .181 | .262 | .179 | .259 | .177 | .258 | .175 | .257 |
| | 192 | .254 | .314 | .248 | .307 | .247 | .307 | .242 | .303 | .241 | .302 | .239 | .300 |
| | 336 | .317 | .353 | .314 | .350 | .315 | .349 | .300 | .340 | .298 | .340 | .298 | .339 |
| | 720 | .412 | .407 | .411 | .405 | .410 | .404 | .401 | .397 | .394 | .398 | .395 | .396 |
| | Avg. | .293 | .337 | .289 | .331 | .288 | .330 | .280 | .326 | .277 | .324 | .277 | .323 |
| PEMS03 | 12 | .071 | .174 | .069 | .170 | .067 | .168 | .080 | .188 | .077 | .187 | .071 | .179 |
| | 24 | .097 | .208 | .080 | .184 | .078 | .181 | .125 | .236 | .120 | .232 | .102 | .216 |
| | 48 | .161 | .272 | .112 | .215 | .108 | .214 | .231 | .324 | .216 | .312 | .176 | .288 |
| | 96 | .240 | .338 | .143 | .246 | .138 | .247 | .383 | .430 | .353 | .405 | .285 | .379 |
| | Avg. | .142 | .248 | .101 | .204 | .098 | .203 | .205 | .294 | .192 | .284 | .159 | .266 |
| PEMS04 | 12 | .081 | .188 | .071 | .175 | .071 | .174 | .097 | .205 | .093 | .202 | .083 | .191 |
| | 24 | .099 | .211 | .079 | .186 | .080 | .187 | .149 | .260 | .138 | .250 | .113 | .226 |
| | 48 | .133 | .246 | .095 | .203 | .093 | .201 | .266 | .355 | .237 | .333 | .172 | .285 |
| | 96 | .172 | .283 | .109 | .220 | .109 | .219 | .432 | .463 | .379 | .430 | .258 | .358 |
| | Avg. | .121 | .232 | .088 | .196 | .088 | .195 | .236 | .321 | .212 | .304 | .156 | .265 |
| PEMS07 | 12 | .067 | .165 | .056 | .151 | .056 | .150 | .074 | .177 | .072 | .175 | .065 | .165 |
| | 24 | .088 | .190 | .076 | .173 | .073 | .169 | .121 | .228 | .116 | .223 | .093 | .198 |
| | 48 | .113 | .218 | .097 | .185 | .096 | .183 | .226 | .316 | .204 | .298 | .144 | .251 |
| | 96 | .172 | .283 | .119 | .202 | .114 | .195 | .379 | .416 | .344 | .385 | .221 | .318 |
| | Avg. | .102 | .205 | .087 | .178 | .085 | .174 | .200 | .284 | .184 | .270 | .131 | .233 |
| PEMS08 | 12 | .088 | .193 | .078 | .181 | .078 | .181 | .096 | .201 | .091 | .196 | .084 | .187 |
| | 24 | .138 | .243 | .109 | .214 | .109 | .214 | .158 | .260 | .142 | .246 | .125 | .231 |
| | 48 | .334 | .353 | .217 | .240 | .196 | .236 | .299 | .368 | .260 | .338 | .204 | .304 |
| | 96 | .458 | .436 | .232 | .257 | .228 | .252 | .555 | .504 | .494 | .451 | .334 | .394 |
| | Avg. | .254 | .306 | .159 | .223 | .153 | .221 | .277 | .333 | .247 | .308 | .187 | .279 |
| Exchange | 96 | .086 | .206 | .086 | .206 | .085 | .205 | .083 | .203 | .084 | .203 | .082 | .200 |
| | 192 | .177 | .299 | .174 | .298 | .173 | .297 | .175 | .299 | .175 | .298 | .173 | .296 |
| | 336 | .338 | .422 | .324 | .412 | .323 | .412 | .325 | .415 | .325 | .413 | .323 | .411 |
| | 720 | .847 | .691 | .824 | .687 | .815 | .675 | .839 | .693 | .835 | .688 | .834 | .687 |
| | Avg. | .368 | .409 | .352 | .401 | .349 | .398 | .356 | .403 | .355 | .400 | .353 | .399 |
| Weather | 96 | .174 | .215 | .162 | .205 | .160 | .204 | .196 | .235 | .166 | .210 | .163 | .209 |
| | 192 | .224 | .258 | .211 | .251 | .210 | .250 | .240 | .271 | .214 | .252 | .210 | .251 |
| | 336 | .281 | .298 | .268 | .293 | .266 | .290 | .291 | .307 | .269 | .292 | .267 | .292 |
| | 720 | .359 | .351 | .346 | .343 | .345 | .341 | .363 | .353 | .346 | .342 | .344 | .342 |
| | Avg. | .260 | .281 | .247 | .273 | .245 | .271 | .272 | .292 | .249 | .274 | .246 | .273 |
| Solar | 96 | .201 | .234 | .197 | .233 | .185 | .222 | .233 | .296 | .217 | .257 | .207 | .252 |
| | 192 | .238 | .261 | .229 | .257 | .221 | .246 | .260 | .316 | .245 | .274 | .239 | .272 |
| | 336 | .248 | .273 | .239 | .269 | .231 | .266 | .276 | .323 | .265 | .287 | .261 | .288 |
| | 720 | .249 | .275 | .246 | .275 | .241 | .268 | .273 | .316 | .265 | .287 | .263 | .292 |
| | Avg. | .234 | .261 | .228 | .258 | .220 | .249 | .261 | .313 | .248 | .276 | .242 | .277 |
| ECL | 96 | .148 | .240 | .133 | .229 | .132 | .228 | .201 | .287 | .164 | .253 | .162 | .252 |
| | 192 | .167 | .258 | .152 | .247 | .150 | .244 | .209 | .297 | .174 | .262 | .173 | .262 |
| | 336 | .179 | .272 | .165 | .262 | .164 | .260 | .228 | .316 | .191 | .279 | .190 | .278 |
| | 720 | .220 | .310 | .191 | .286 | .187 | .280 | .273 | .350 | .232 | .312 | .230 | .312 |
| | Avg. | .179 | .270 | .161 | .256 | .158 | .256 | .228 | .313 | .190 | .277 | .189 | .276 |
| Average | | .275 | .318 | .244 | .297 | .241 | .295 | .297 | .346 | .280 | .330 | .262 | .319 |
| 1st Count | | 0 | 0 | 9 | 9 | 46 | 46 | 0 | 0 | 4 | 7 | 44 | 46 |
| 2nd Count | | 10 | 9 | 39 | 39 | 3 | 2 | 4 | 7 | 43 | 41 | 4 | 2 |

Table M.1: **TS backbones w/o CID ability.** Full results of TS forecasting tasks.

| Models | | S-Mamba | | + CN | | + ACN | | TSMixer | | + CN | | + ACN | |
|---|---|---|---|---|---|---|---|---|---|---|---|---|---|---|
| Metric | | MSE | MAE | MSE | MAE | MSE | MAE | MSE | MAE | MSE | MAE | MSE | MAE |
| ETTh1 | 96 | .385 | .404 | .385 | .405 | .381 | .403 | .398 | .411 | .380 | .399 | .389 | .402 |
| | 192 | .445 | .441 | .442 | .438 | .439 | .435 | .452 | .441 | .430 | .426 | .441 | .426 |
| | 336 | .491 | .462 | .491 | .465 | .480 | .459 | .495 | .462 | .473 | .448 | .476 | .454 |
| | 720 | .506 | .497 | .501 | .492 | .492 | .488 | .501 | .482 | .470 | .466 | .496 | .479 |
| | Avg. | .457 | .452 | .455 | .450 | .448 | .446 | .462 | .449 | .438 | .435 | .453 | .441 |
| ETTh2 | 96 | .297 | .349 | .290 | .342 | .289 | .342 | .316 | .358 | .308 | .354 | .311 | .353 |
| | 192 | .378 | .399 | .371 | .394 | .370 | .393 | .401 | .409 | .378 | .399 | .399 | .399 |
| | 336 | .425 | .435 | .418 | .429 | .415 | .425 | .440 | .444 | .428 | .436 | .416 | .428 |
| | 720 | .432 | .448 | .422 | .441 | .423 | .441 | .454 | .462 | .433 | .449 | .426 | .443 |
| | Avg. | .383 | .408 | .375 | .401 | .374 | .400 | .403 | .418 | .387 | .410 | .386 | .407 |
| ETTm1 | 96 | .326 | .368 | .328 | .365 | .326 | .363 | .330 | .366 | .319 | .358 | .316 | .355 |
| | 192 | .378 | .393 | .375 | .392 | .374 | .391 | .374 | .391 | .364 | .385 | .363 | .382 |
| | 336 | .410 | .414 | .409 | .415 | .406 | .412 | .417 | .415 | .394 | .404 | .394 | .409 |
| | 720 | .474 | .451 | .474 | .451 | .472 | .448 | .484 | .450 | .466 | .444 | .466 | .444 |
| | Avg. | .398 | .407 | .397 | .406 | .394 | .404 | .401 | .406 | .386 | .398 | .385 | .397 |
| ETTm2 | 96 | .182 | .266 | .176 | .260 | .175 | .259 | .178 | .261 | .177 | .261 | .175 | .257 |
| | 192 | .252 | .313 | .246 | .307 | .243 | .303 | .245 | .305 | .248 | .308 | .239 | .301 |
| | 336 | .313 | .349 | .311 | .348 | .308 | .345 | .313 | .348 | .311 | .346 | .303 | .342 |
| | 720 | .416 | .409 | .409 | .403 | .409 | .405 | .416 | .406 | .410 | .403 | .401 | .400 |
| | Avg. | .290 | .333 | .286 | .329 | .284 | .328 | .287 | .330 | .286 | .329 | .280 | .325 |
| PEMS03 | 12 | .066 | .171 | .062 | .164 | .062 | .164 | .066 | .171 | .065 | .169 | .064 | .169 |
| | 24 | .088 | .197 | .080 | .185 | .079 | .185 | .090 | .202 | .089 | .197 | .085 | .195 |
| | 48 | .165 | .277 | .121 | .231 | .120 | .230 | .142 | .253 | .137 | .244 | .133 | .245 |
| | 96 | .213 | .313 | .170 | .276 | .168 | .275 | .218 | .319 | .204 | .300 | .196 | .312 |
| | Avg. | .133 | .240 | .108 | .214 | .107 | .213 | .129 | .236 | .124 | .228 | .120 | .230 |
| PEMS04 | 12 | .073 | .177 | .069 | .170 | .072 | .175 | .074 | .181 | .074 | .179 | .072 | .176 |
| | 24 | .084 | .192 | .077 | .182 | .085 | .191 | .091 | .200 | .091 | .200 | .087 | .197 |
| | 48 | .101 | .213 | .091 | .196 | .102 | .212 | .121 | .239 | .121 | .234 | .117 | .234 |
| | 96 | .125 | .236 | .103 | .210 | .124 | .231 | .173 | .294 | .168 | .274 | .159 | .280 |
| | Avg. | .096 | .205 | .085 | .189 | .095 | .202 | .115 | .228 | .114 | .222 | .109 | .222 |
| PEMS07 | 12 | .060 | .157 | .054 | .145 | .054 | .147 | .066 | .167 | .063 | .161 | .058 | .155 |
| | 24 | .082 | .184 | .068 | .160 | .065 | .160 | .088 | .190 | .087 | .187 | .079 | .179 |
| | 48 | .100 | .204 | .084 | .175 | .080 | .179 | .125 | .220 | .127 | .224 | .113 | .215 |
| | 96 | .117 | .218 | .105 | .189 | .094 | .188 | .181 | .273 | .184 | .265 | .161 | .264 |
| | Avg. | .090 | .191 | .078 | .168 | .073 | .167 | .115 | .210 | .115 | .209 | .103 | .203 |
| PEMS08 | 12 | .076 | .178 | .071 | .169 | .071 | .171 | .081 | .186 | .080 | .182 | .079 | .181 |
| | 24 | .110 | .216 | .093 | .192 | .092 | .195 | .115 | .222 | .113 | .217 | .110 | .214 |
| | 48 | .173 | .254 | .134 | .227 | .133 | .232 | .188 | .289 | .181 | .274 | .179 | .277 |
| | 96 | .271 | .321 | .233 | .277 | .190 | .266 | .362 | .402 | .295 | .327 | .304 | .360 |
| | Avg. | .157 | .242 | .133 | .216 | .121 | .216 | .186 | .275 | .167 | .250 | .167 | .258 |
| Exchange | 96 | .086 | .206 | .086 | .206 | .086 | .205 | .086 | .205 | .085 | .203 | .084 | .203 |
| | 192 | .181 | .303 | .180 | .302 | .179 | .302 | .177 | .302 | .175 | .298 | .173 | .297 |
| | 336 | .331 | .417 | .323 | .411 | .324 | .412 | .329 | .414 | .321 | .411 | .317 | .408 |
| | 720 | .858 | .699 | .860 | .699 | .841 | .690 | .868 | .704 | .851 | .697 | .846 | .694 |
| | Avg. | .364 | .407 | .362 | .405 | .357 | .402 | .365 | .406 | .358 | .402 | .356 | .400 |
| Weather | 96 | .165 | .209 | .160 | .205 | .162 | .207 | .181 | .228 | .159 | .206 | .156 | .204 |
| | 192 | .215 | .255 | .208 | .250 | .209 | .251 | .227 | .263 | .209 | .252 | .206 | .250 |
| | 336 | .273 | .296 | .268 | .292 | .268 | .294 | .280 | .300 | .267 | .295 | .263 | .293 |
| | 720 | .353 | .349 | .348 | .344 | .350 | .348 | .353 | .347 | .350 | .345 | .343 | .343 |
| | Avg. | .252 | .277 | .246 | .273 | .247 | .274 | .260 | .285 | .246 | .274 | .242 | .272 |
| Solar | 96 | .207 | .246 | .194 | .230 | .189 | .229 | .222 | .281 | .200 | .231 | .215 | .251 |
| | 192 | .240 | .272 | .227 | .258 | .223 | .258 | .261 | .301 | .251 | .265 | .250 | .277 |
| | 336 | .262 | .286 | .248 | .277 | .246 | .278 | .271 | .299 | .269 | .278 | .264 | .288 |
| | 720 | .267 | .293 | .250 | .282 | .252 | .285 | .267 | .293 | .266 | .292 | .254 | .282 |
| | Avg. | .244 | .275 | .230 | .262 | .228 | .261 | .255 | .294 | .246 | .267 | .245 | .274 |
| ECL | 96 | .139 | .237 | .135 | .233 | .135 | .233 | .177 | .278 | .147 | .250 | .146 | .248 |
| | 192 | .165 | .261 | .157 | .255 | .155 | .250 | .193 | .293 | .166 | .266 | .162 | .262 |
| | 336 | .177 | .274 | .168 | .267 | .162 | .268 | .215 | .315 | .187 | .288 | .177 | .278 |
| | 720 | .214 | .304 | .190 | .289 | .186 | .286 | .260 | .352 | .223 | .316 | .209 | .304 |
| | Avg. | .174 | .269 | .163 | .261 | .162 | .259 | .211 | .310 | .181 | .280 | .174 | .273 |
| Average | | .253 | .309 | .243 | .298 | .240 | .297 | .266 | .321 | .254 | .309 | .243 | .308 |
| 1st Count | | 1 | 0 | 15 | 25 | 38 | 31 | 0 | 0 | 10 | 16 | 40 | 36 |
| 2nd Count | | 10 | 14 | 31 | 23 | 9 | 17 | 9 | 6 | 35 | 30 | 8 | 12 |

Table M.2: **TS backbones w/ CID ability.** Full results of TS forecasting tasks.

# N. Full Results: Application of PCN

## N.1. Application of PCN to non-TSFMs

Although PCN is developed for scenarios with multiple datasets and varying C (e.g., TSFM), it can also be applied to single-task models trained on a single dataset, assuming the number of channels remains unknown. Table N.1 presents the results of applying PCN with $K = 5$ to iTransformer. The results, averaged over 12 datasets and 4 horizons, show an 8.4% performance improvement, which is smaller than the improvement achieved by CN and ACN.

| iTrans. | Average MSE across 4 horizons | | | | | | | | | | | | Avg. | Imp. |
|---------|-------|-------|-------|-------|--------|--------|--------|--------|----------|---------|-------|------|------|------|
|         | ETTh1 | ETTh2 | ETTm1 | ETTm2 | PEMS03 | PEMS04 | PEMS07 | PEMS08 | Exchange | Weather | Solar | ECL  |      |      |
| -       | .457  | .384  | .408  | .293  | .142   | .121   | .102   | .254   | .368     | .260    | .234  | .179 | .275 | -    |
| CN      | .441  | .376  | .396  | .289  | .101   | **.088** | .087 | .159   | .352     | .247    | .228  | .161 | .244 | 11.3% |
| ACN     | .438  | **.374** | **.395** | **.288** | **.098** | **.088** | **.085** | **.153** | .349 | **.245** | **.220** | **.158** | **.241** | **12.4%** |
| PCN     | **.437** | .376 | .404 | .289  | .117   | .120   | .101   | .176   | **.349** | .257   | .232  | .169 | .252 | 8.4% |

Table N.1: Application of PCN to iTransformer.

## N.2. Application of PCN to TSFMs

Table 4 summarizes the results of 20 forecasting and 18 classification tasks under supervised and prompt-tuning settings, with full results for both tasks shown in Table N.2 and Table N.3, respectively.

| UniTS (LN) | | Supervised | | | | Prompt-Tuning | | | |
|------------|-----|-----|-----|-----|-----|-----|-----|-----|-----|
|            |     | - | | + PCN | | - | | + PCN | |
| Dataset    | $H$ | MSE | MAE | MSE | MAE | MSE | MAE | MSE | MAE |
| NN5        | 112 | .635 | .556 | **.610** | **.545** | .611 | .552 | **.602** | **.543** |
| ECL        | 96  | .172 | .273 | **.168** | **.272** | .174 | **.277** | **.173** | .278 |
|            | 192 | .185 | .284 | **.182** | **.283** | **.189** | **.289** | **.189** | .292 |
|            | 336 | **.196** | .297 | .197 | **.296** | **.205** | **.304** | **.205** | .306 |
|            | 720 | .238 | **.321** | **.227** | **.321** | .251 | .340 | **.241** | **.334** |
| ETTh1      | 96  | .390 | .408 | **.388** | **.406** | .390 | .411 | **.384** | **.405** |
|            | 192 | **.428** | **.432** | .438 | .434 | **.432** | .439 | .433 | **.432** |
|            | 336 | **.462** | **.451** | .477 | .454 | .480 | .460 | **.472** | **.450** |
|            | 720 | .489 | .476 | **.484** | **.475** | .532 | .500 | **.492** | **.475** |
| Exchange   | 192 | .239 | .342 | **.202** | **.323** | .221 | .337 | **.207** | **.329** |
|            | 336 | .479 | .486 | **.383** | **.446** | .387 | .453 | **.366** | **.441** |
| ILI        | 60  | 2.48 | .944 | **1.93** | **.895** | 2.45 | .994 | **2.14** | **.940** |
| Traffic    | 96  | .496 | .325 | **.483** | **.320** | .502 | .330 | **.481** | **.318** |
|            | 192 | .497 | .327 | **.495** | **.324** | .523 | .331 | **.505** | **.322** |
|            | 336 | .509 | .328 | **.506** | **.326** | .552 | .338 | **.535** | **.330** |
|            | 720 | **.525** | .350 | .536 | **.341** | .626 | .369 | **.591** | **.352** |
| Weather    | 96  | .161 | .211 | **.157** | **.207** | .175 | **.214** | **.166** | .217 |
|            | 192 | .212 | .255 | **.205** | **.251** | .226 | .266 | **.219** | **.261** |
|            | 336 | .266 | .295 | **.262** | **.293** | .280 | .303 | **.275** | **.299** |
|            | 720 | .343 | .344 | **.338** | **.342** | .352 | .350 | **.350** | **.348** |
| Best Count (/20) | | 4 | 3 | **16** | **18** | 3 | 4 | **20** | 16 |
| Average    |     | .469 | .386 | **.433** | **.378** | .478 | .393 | **.453** | **.384** |

Table N.2: Results of multi-task forecasting with UniTS.

| UniTS (LN) | Supervised | | Prompt-Tuning | |
|---|---|---|---|---|
| | - | + PCN | - | + PCN |
| Heartbeat | 59.0 | **71.7** | 69.3 | **73.1** |
| JapaneseVowels | **93.5** | 92.7 | 90.8 | **92.7** |
| PEMS-SF | 83.2 | **84.9** | **85.0** | 82.7 |
| SelfRegulationSCP2 | 47.8 | **55.0** | **53.3** | 51.7 |
| SpokenArabicDigits | 97.5 | **98.0** | 92.0 | **94.9** |
| UWaveGestureLibrary | 79.1 | **85.3** | 75.6 | **84.1** |
| ECG5000 | 92.6 | **93.6** | 93.4 | **94.0** |
| NonInvasive. | **90.5** | 89.7 | 27.1 | **54.8** |
| Blink | 99.1 | **99.8** | 91.1 | **98.0** |
| FaceDetection | 64.1 | **66.7** | 57.6 | **60.7** |
| ElectricDevices | 60.3 | **62.1** | 55.4 | **59.4** |
| Trace | 91.0 | **96.0** | 82.0 | **92.0** |
| FordB | 76.0 | **76.5** | 62.8 | **67.2** |
| MotionSenseHAR | 92.8 | **93.2** | 93.2 | **94.7** |
| EMOPain | 78.0 | **79.2** | 80.3 | **85.1** |
| Chinatown | 97.7 | **98.0** | 98.0 | **98.3** |
| MelbournePedestrian | 87.3 | **88.2** | 77.0 | **78.5** |
| SharePriceIncrease | 61.9 | **63.1** | **68.4** | **68.4** |
| Best Count (/18) | 2 | **16** | 3 | **16** |
| Average Score | 80.6 | **83.0** | 75.1 | **79.5** |

Table N.3: Results of multi-task classification with UniTS.

# O. Visualization of Forecasting Results

To validate the effectiveness of our method, we visualize the predicted results for various $L$ and $H$ across different backbone architectures and four datasets from diverse domains: ETTm1 (Zhou et al., 2021), Weather (Wu et al., 2021), ECL (Wu et al., 2021), and PEMS (Liu et al., 2022), using three types of normalizations: base (LN), CN, and ACN.

## O.1. Visualization of TSF with iTransformer

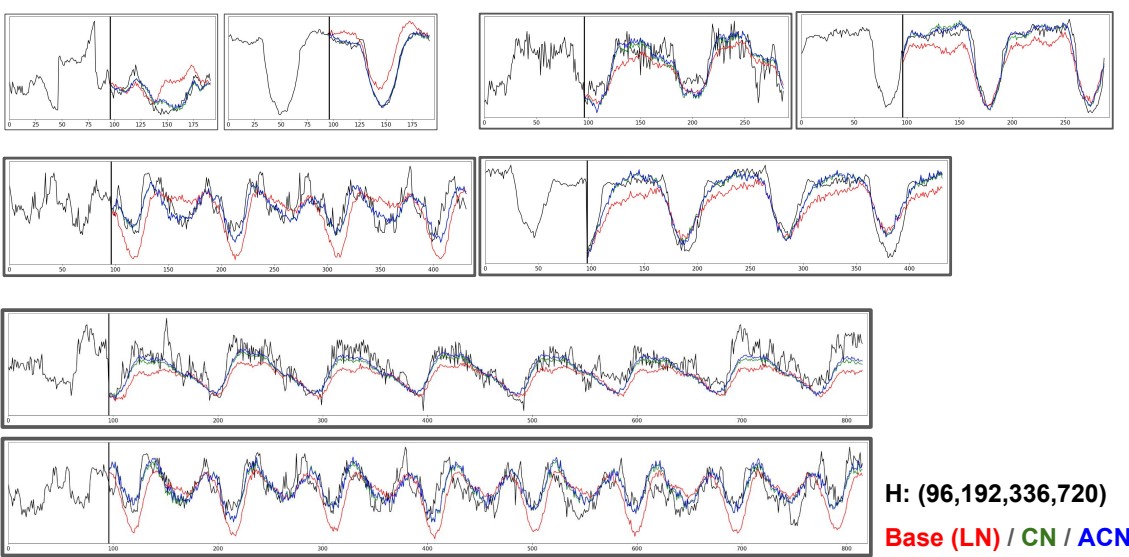

Figure O.1: TS forecasting results of **ETTm1** with **iTransformer**.

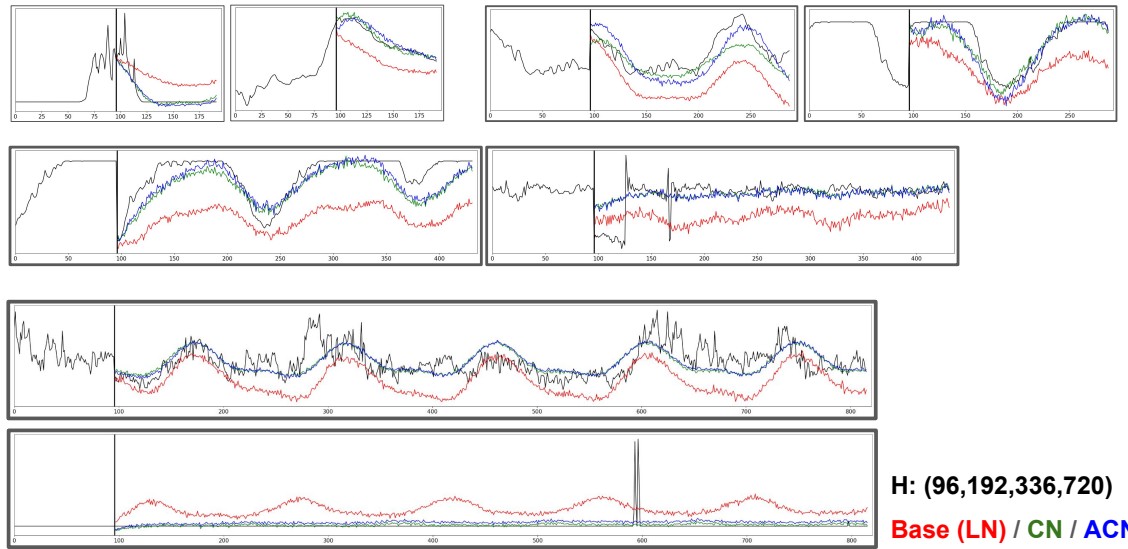

Figure O.2: TS forecasting results of **Weather** with **iTransformer**.

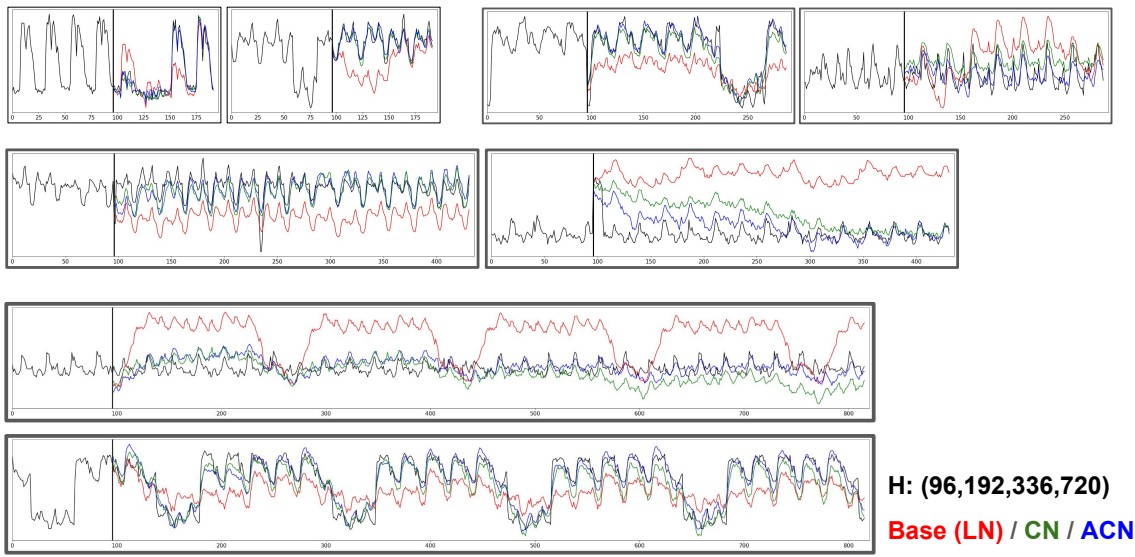

**H: (96,192,336,720)**

**Base (LN) / CN / ACN**

Figure O.3: TS forecasting results of **ECL** with **iTransformer**.

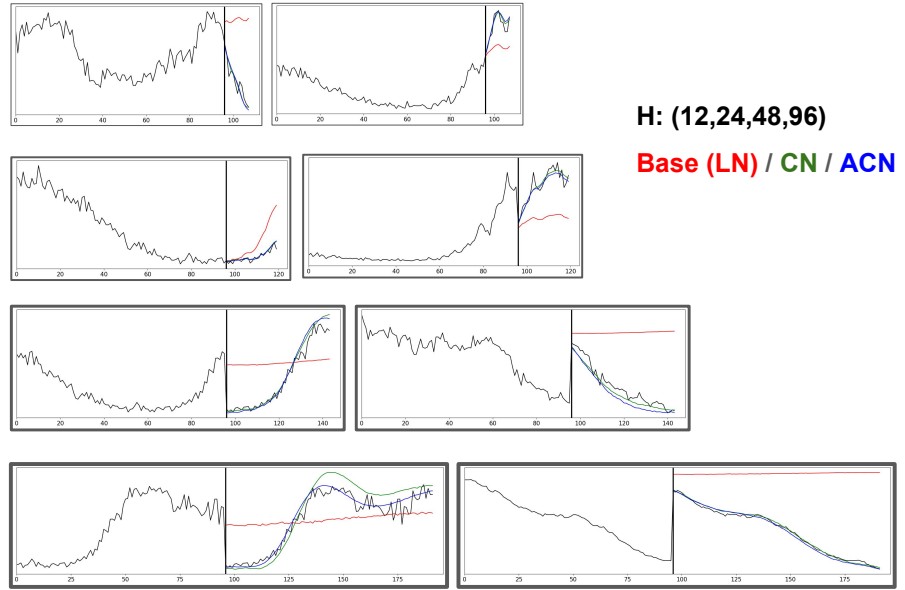

**H: (12,24,48,96)**

**Base (LN) / CN / ACN**

Figure O.4: TS forecasting results of **PEMS07** with **iTransformer**.

## O.2. Visualization of TSF with RMLP

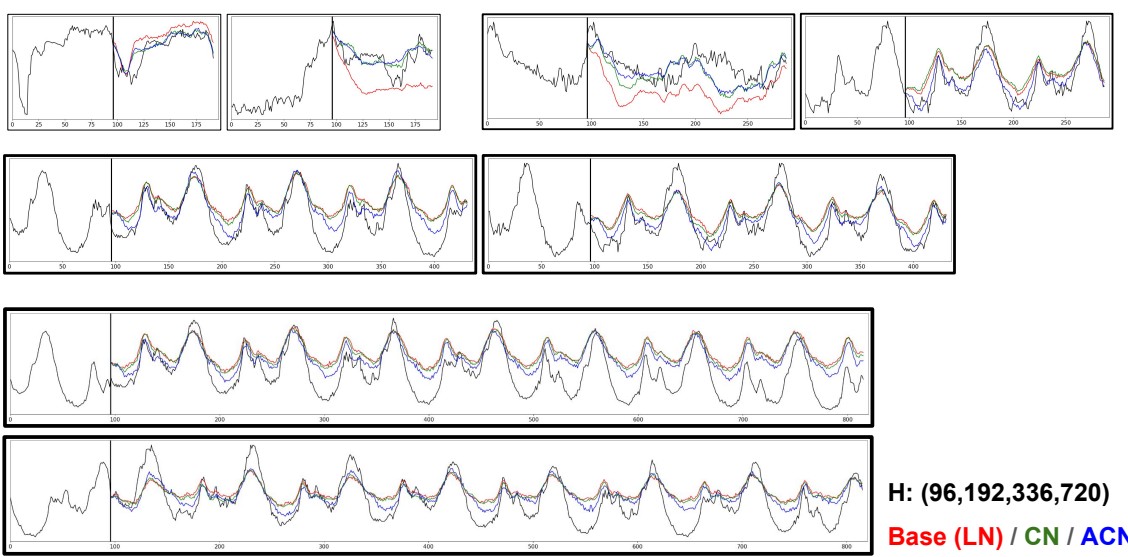

**H: (96,192,336,720)**
**Base (LN)** / **CN** / **ACN**

Figure O.5: TS forecasting results of **ETTm1** with **RMLP**.

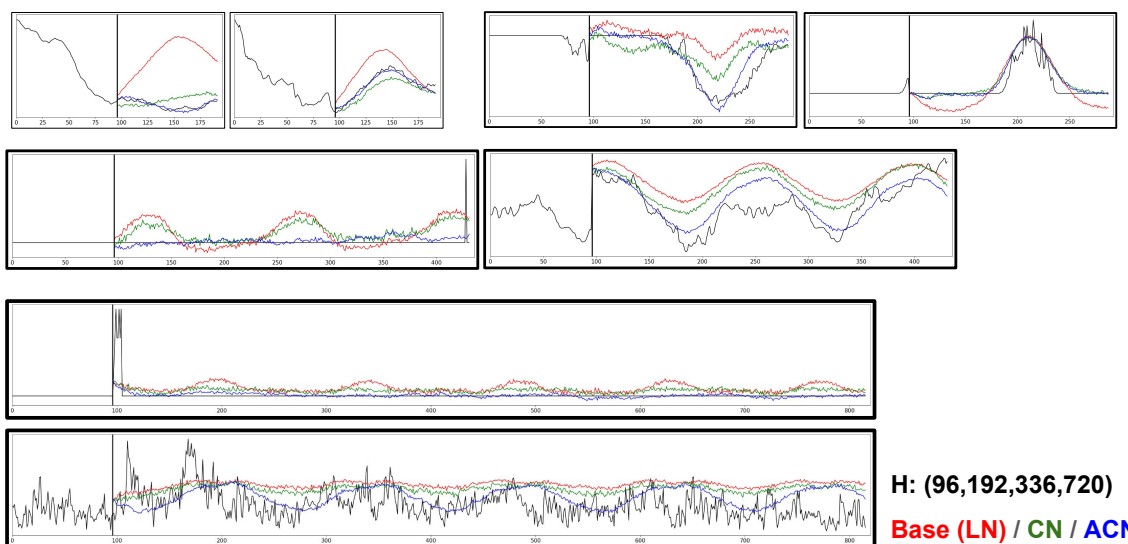

**H: (96,192,336,720)**
**Base (LN)** / **CN** / **ACN**

Figure O.6: TS forecasting results of **Weather** with **RMLP**.

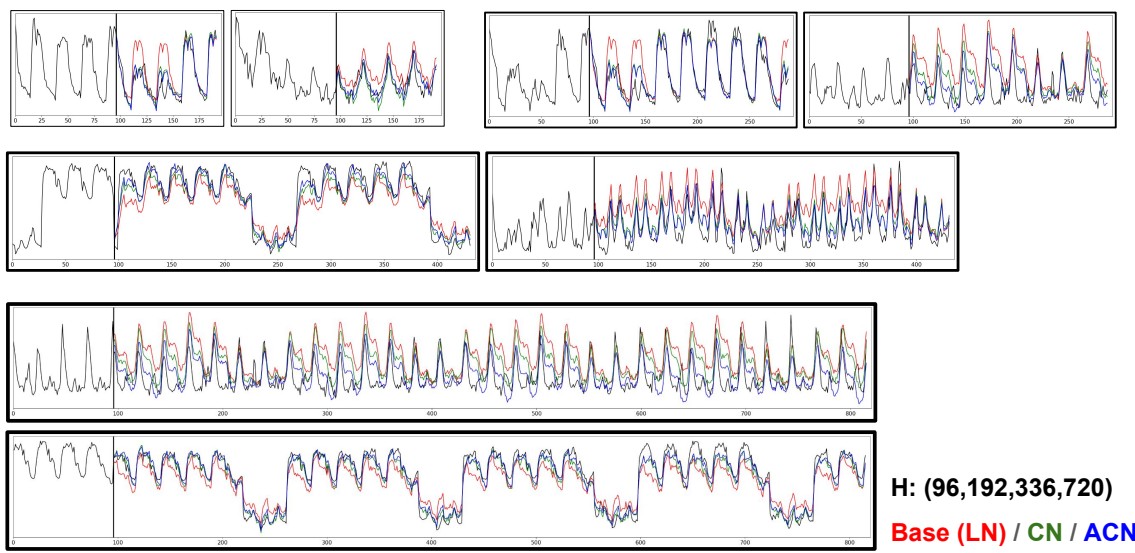

Figure O.7: TS forecasting results of **ECL** with **RMLP**.

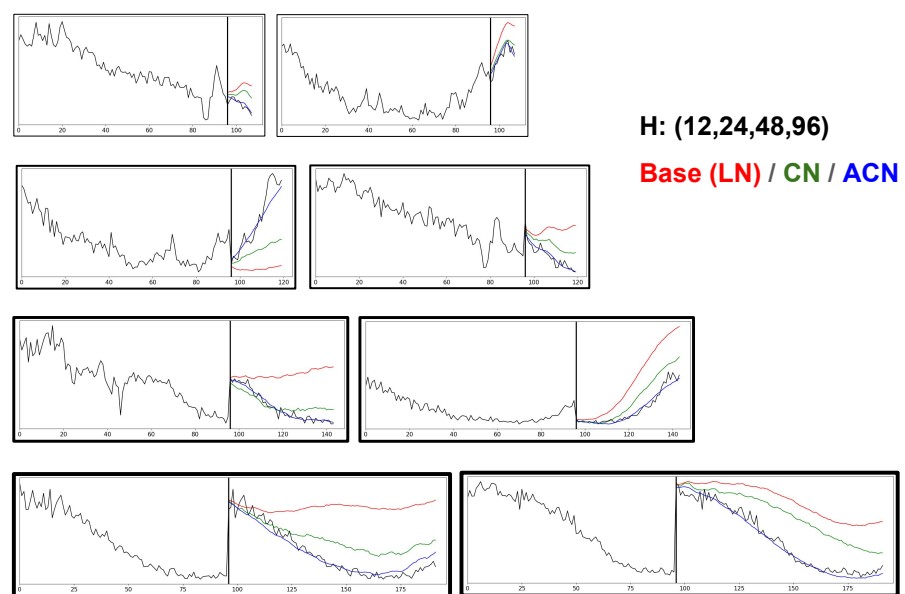

Figure O.8: TS forecasting results of **PEMS07** with **RMLP**.

## O.3. Visualization of TSF with TSMixer

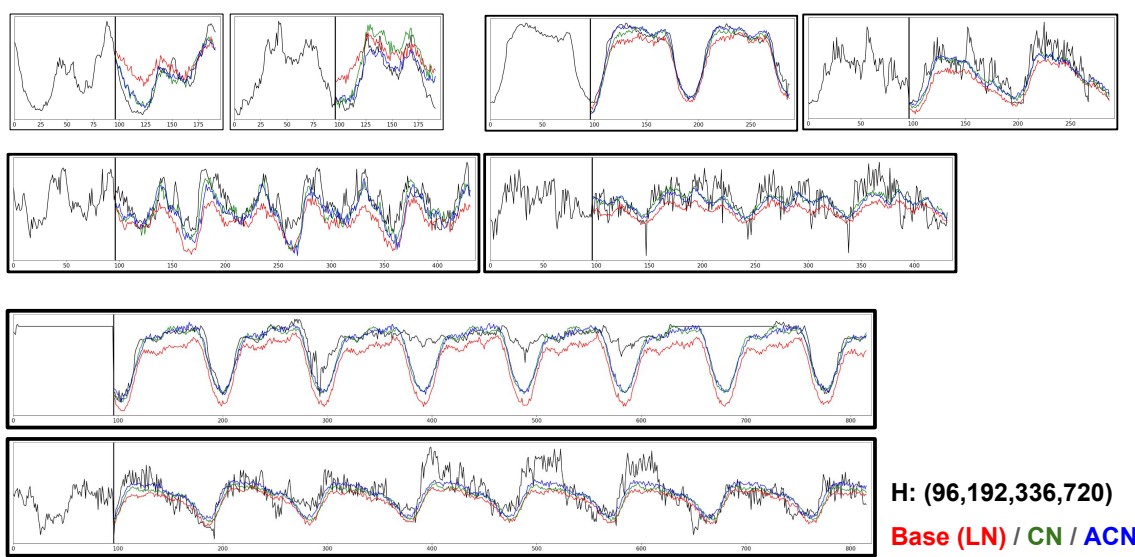

Figure O.9: TS forecasting results of **ETTm1** with **TSMixer**.

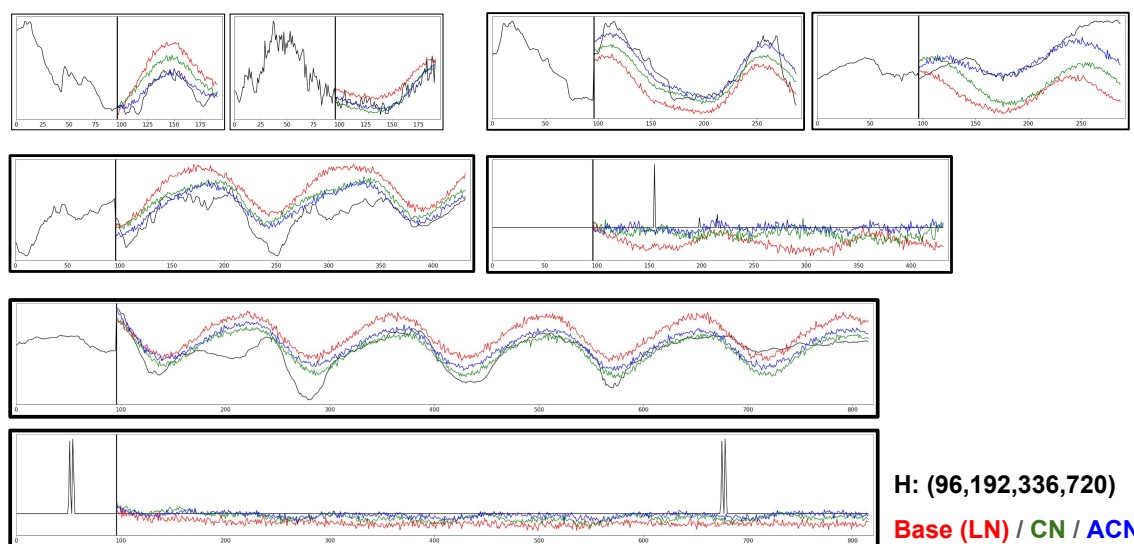

Figure O.10: TS forecasting results of **Weather** with **TSMixer**.

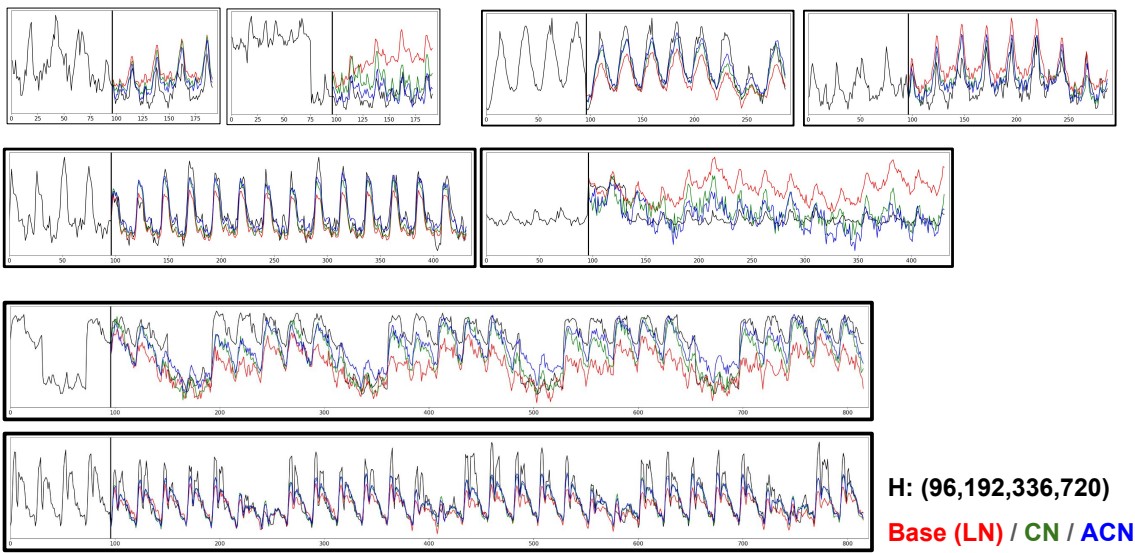

H: (96,192,336,720)

Base (LN) / CN / ACN

Figure O.11: TS forecasting results of **ECL** with **TSMixer**.

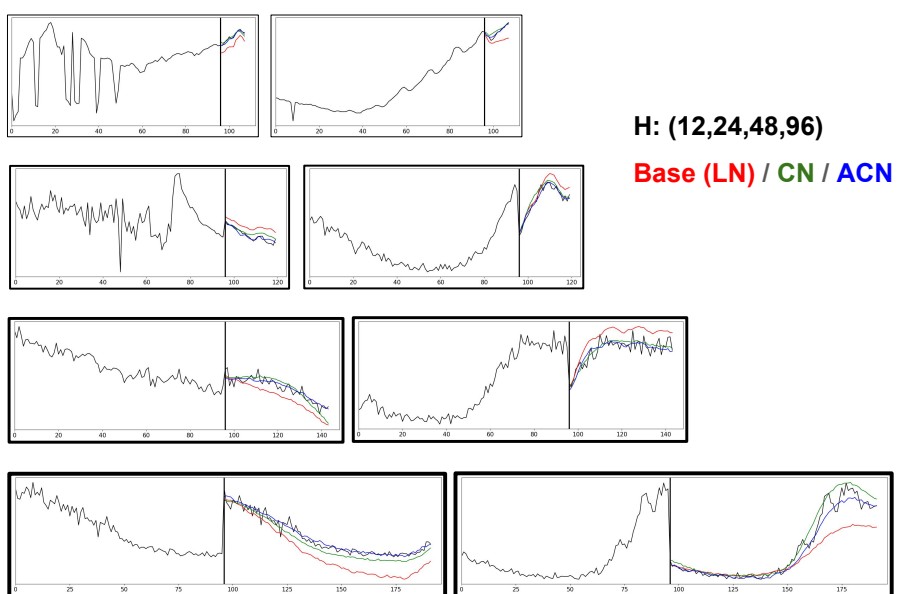

H: (12,24,48,96)

Base (LN) / CN / ACN

Figure O.12: TS forecasting results of **PEMS07** with **TSMixer**.

## O.4. Visualization of TSF with S-Mamba

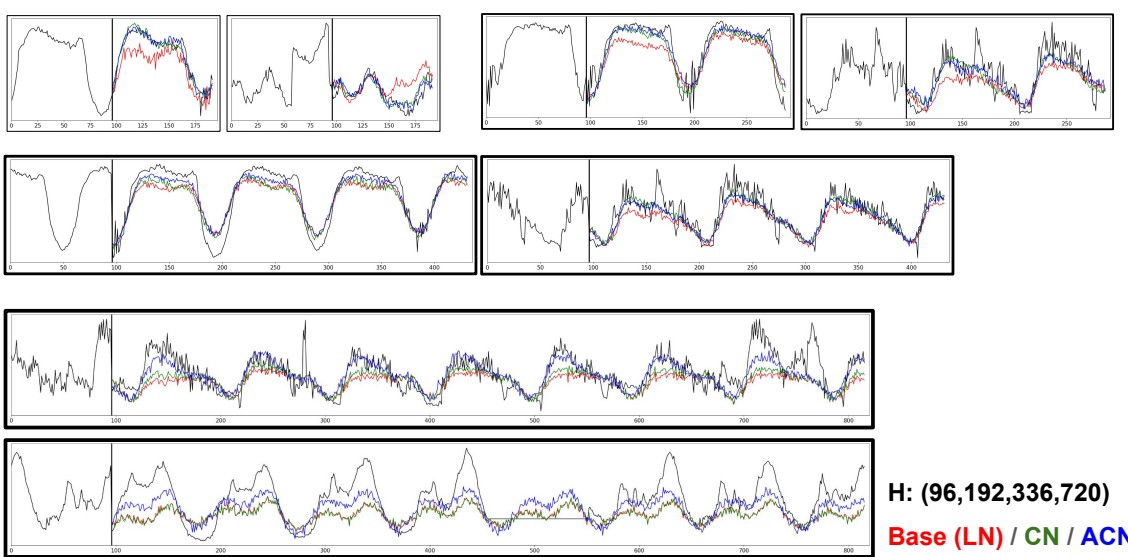

Figure O.13: TS forecasting results of **ETTm1** with **S-Mamba**.

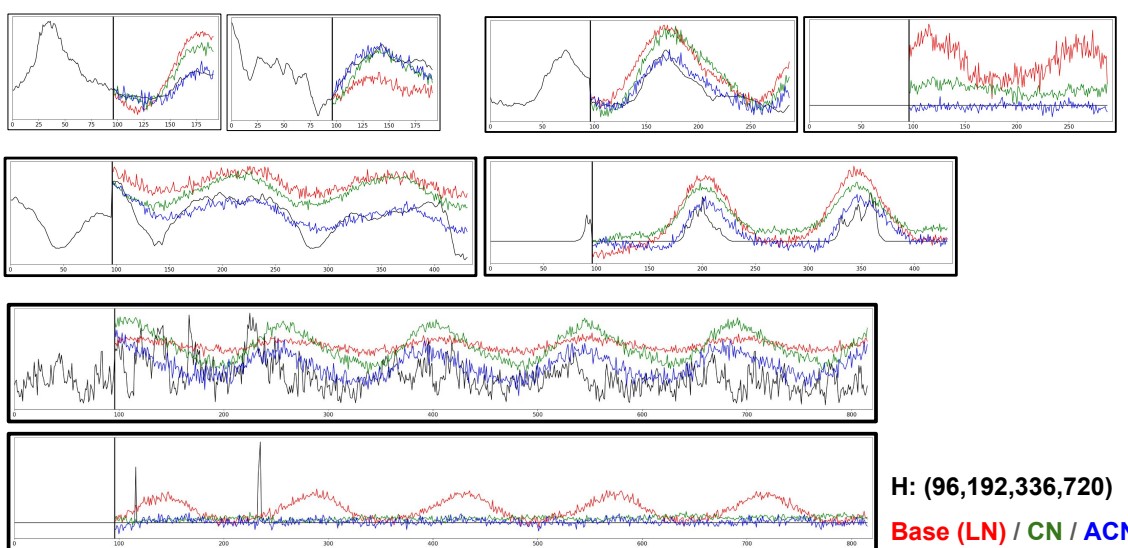

Figure O.14: TS forecasting results of **Weather** with **S-Mamba**.

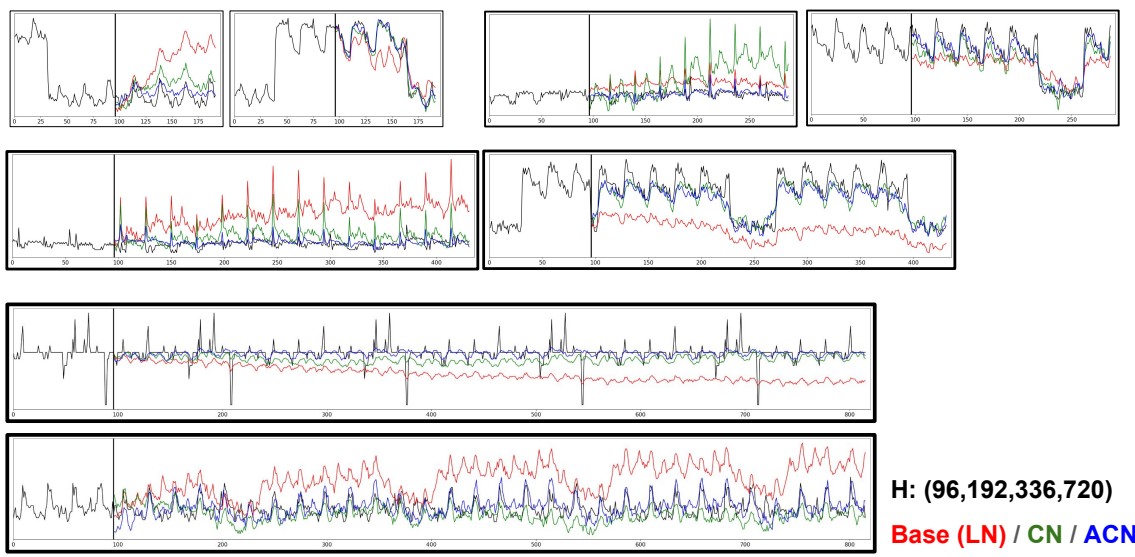

**H: (96,192,336,720)**

**Base (LN) / CN / ACN**

Figure O.15: TS forecasting results of **ECL** with **S-Mamba**.

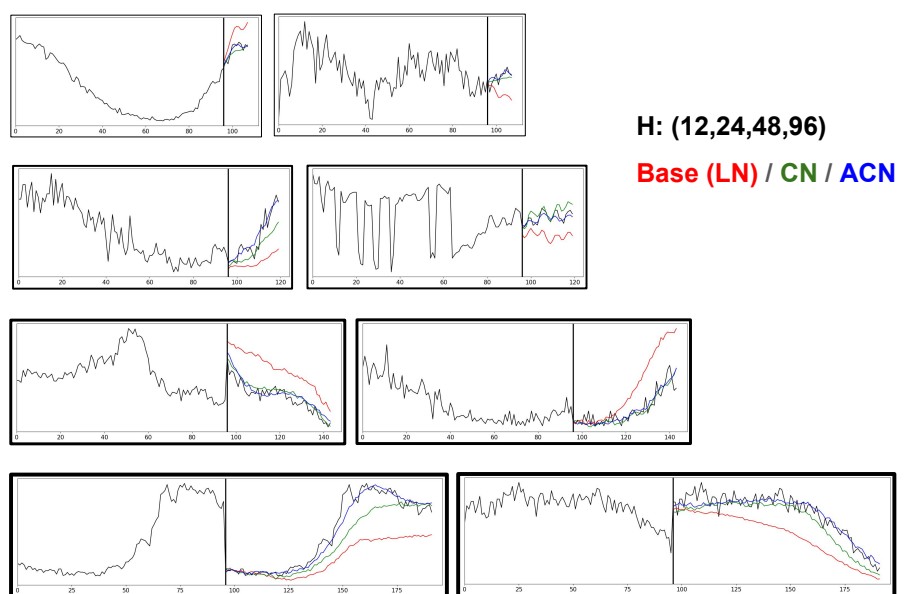

**H: (12,24,48,96)**

**Base (LN) / CN / ACN**

Figure O.16: TS forecasting results of **PEMS07** with **S-Mamba**.

