# OpenReview forum: "Channel Normalization for Time Series Channel Identification"
_ICML.cc/2025/Conference — ICML 2025 poster_

### Official Review · Reviewer_LY88 · 2025-02-23

**Overall Recommendation:** 3

**Summary:**

The paper talk about the importance of Channel Identifiability (CID) when modeling multivariate time series data. The paper talk about how existing methods failed to provide CID capability. To solve this problem, the model proposed various Channel Normalization (CN) method. CN is a type of normalization method that uses different affine transformation for different channel. In additional a simple CN method, the paper also provided two extension 1) the Adaptive CN (ACN) method which also models the dependency between input dimensions in a data dependent fashion and 2) Prototypical CN (PCN) method which uses learned prototype to model different dimensions.

**Claims And Evidence:**

- Claims:
    1. Some existing methods lack CID capability, potentially leading to suboptimal performance.
    2. CN is a simple yet effective method to provide CID capability for different models.
    3. ACN improves upon CN by performing normalization in a data-dependent fashion.
    4. PCN also provides CID capability.
- Evidence:
    1. This claim is supported by Figure 1 and Table 1.
    2. This claim is backed by experimental results and can be easily inferred from the methodology, as the method is straightforward.
    3. This claim is also supported by experimental results and can be easily inferred from the methodology.
    4. This claim is not fully substantiated. Although PCN improves model performance, the source of the improvement is unclear. Moreover, I am not fully convinced that PCN provides CID, which I will elaborate on in later sections.
- Missing: Theoretical analysis is absent, but it is not necessary as it is easy to see how CN/ACN provides CID.

**Essential References Not Discussed:**

To the best of my knowledge, there are no essential references missing from the paper's discussion.

**Experimental Designs Or Analyses:**

Yes, I have checked the experimental designs and analyses. No severe issues were found.

**Methods And Evaluation Criteria:**

Yes, the proposed methods and evaluation criteria make sense for the problem at hand.

**Other Comments Or Suggestions:**

The variables used in Section 4 are not introduced. What are B, C, D, and K? It is difficult to understand the methods without looking at the source code.

**Other Strengths And Weaknesses:**

- Strengths: The motivation behind the proposed methods is clear, and the proposed methods are simple yet effective.
- Weaknesses: One of the proposed methods (PCN) seems out of place and may not fully align with the paper's main contributions.

**Questions For Authors:**

1. How does PCN provide CID?

**Relation To Broader Scientific Literature:**

The proposed CN methods are simple yet effective, which is a novel contribution not seen in prior work.

**Theoretical Claims:**

I am not fully convinced that PCN provides CID capability. Comparing the algorithms for CN (Algorithm 1), ACN (Algorithm 2), and PCN (Algorithm 3), we can see that PCN is the only one without a global parameter for normalization. Without a global parameter, the normalization process cannot differentiate between two channels if they have identical inputs (Figure 1). This could be the main reason why PCN performs worse than CN and ACN. The paper would be stronger if PCN were not included in the main text. I believe PCN is more of an extension of CN (i.e., an effective channel normalization method that does not provide CID) and should be discussed in the appendix.

---

> ### Author Rebuttal · Authors · 2025-03-31
>
> ## Weakness 1. PCN’s CID capability
> > Reviewer: *I am not fully convinced that PCN provides CID capability. ~ I believe PCN is more of an extension of CN (i.e., an effective channel normalization method that does not provide CID) and should be discussed in the appendix.*
>
> Thank you for pointing this out. We acknowledge that PCN indeed does not provide CID capability, as PCN does not assign an identity to **“individual channels”** but rather to **“channel clusters”**.
>
> PCN is specifically designed for scenarios encountering **new** channels in **new** datasets (common settings for foundation models), where ***channel identification becomes infeasible***. Our proposed PCN address this challenge by assigning **“channel cluster”** identification via learnable prototypes as affine transformation parameters after normalization.
>
> While the current manuscript presents PCN as an extension of CN in that it also provides **a form of identity (to either channels or channel clusters)**, we recognize the reviewer's concern that it may require a relaxed version of CID for channel clusters. We will clarify this distinction in the revision.
>
> &nbsp;
>
> ## Weakness 2. Missing notations
> > Reviewer: *The variables used in Section 4 are not introduced. What are $B$, $C$, $D$, and $K$? It is difficult to understand the methods without looking at the source code.*
>
> Thank you for pointing this out. We found some of these definitions were missing.
>
> $B$ represents the batch size, $C$ denotes the number of channels, $D$ is the hidden dimension, and $K$ refers to the number of prototypes.
>
> To improve clarity, we will explicitly define $B$, $C$, $D$, and $K$ in **Section 4** and also include them in the algorithm pseudocode.
>
> &nbsp;
>
> &nbsp;
>
> **If there are any unresolved issues, please feel free to discuss them with us!**

---

> > ### Comment · Reviewer_LY88 · 2025-04-02
> >
> > concerns resolved

---

### Official Review · Reviewer_PycD · 2025-03-04

**Overall Recommendation:** 5

**Summary:**

- The Channel Normalization (CN) strategy is proposed to enhance the Channel Identifiability (CID) of Time Series (TS) models by assigning specific parameters to each channel.
- Two variants of CN, Adaptive CN (ACN) and Prototypical CN (PCN), are introduced to dynamically adjust parameters and handle datasets with unknown or varying numbers of channels, respectively.

**Claims And Evidence:**

The paper proves the effectiveness of CN, ACN, and PCN in improving model performance through experiments on multiple models and datasets, such as 12 datasets and 4 backbone networks, using Mean Squared Error (MSE) and Mean Absolute Error (MAE) as indicators. The evidence is relatively sufficient.

**Essential References Not Discussed:**

There are no obvious unreferenced papers.

**Experimental Designs Or Analyses:**

The experiments select multiple datasets and different backbone networks, and compare CN and its variants with other methods. The experimental design is relatively comprehensive.

**Methods And Evaluation Criteria:**

- The proposed methods of CN, ACN, and PCN are designed to address the channel identifiability problem in TS models, with clear improvement goals and reasonable method designs.
- The selection of common TS datasets and evaluation metrics, MSE and MAE, can effectively measure the performance of models in TS forecasting tasks.

**Other Comments Or Suggestions:**

"Figure 1: Channel identifiability. Applying the proposed methods to non-CID models enables to distinguish among channels, producing different outputs (green) even with same inputs (yellow)." Maybe the colors are mispositioned.

**Other Strengths And Weaknesses:**

None.

**Questions For Authors:**

- Q1: I have a concern about the performance of CN, ACN, PCN. The authors should provide some details about the performance.
- Q2: How do authors initialize the parameters of CN, ACN, PCN? How do the different initialization methods affect the metrics?

**Relation To Broader Scientific Literature:**

Based on the literature related to TS forecasting models and normalization methods, the paper proposes a new method to enhance CID. Compared with recent similar studies (such as InjectTST, C - LoRA, etc.), it highlights the advantages and innovativeness of its own methods.

**Theoretical Claims:**

The paper proves that CN can obtain more informative representations and potentially reduce forecasting errors from the perspective of theoretical entropy analysis. Its universality in practical applications remains to be further verified.

---

> ### Author Rebuttal · Authors · 2025-03-31
>
> ## Weakness 1. Miscolored Figure 1
> > Reviewer: *"Figure 1: ~ producing different outputs (green) even with same inputs (yellow)." Maybe the colors are mispositioned.*
>
> Thank you for pointing that out. We will fix it in the revised version.
>
> &nbsp;
>
> &nbsp;
>
> ## Question 1. Details about the performance of CN, ACN, PCN
> > Reviewer: *I have a concern about the performance of CN, ACN, PCN. The authors should provide some details about the performance.*
>
> **[PCN vs. CN/ACN]**
> If your concern is the comparison between PCN and CN/ACN, please refer to **Table 5**. Although PCN extends CN, it underperforms CN in the **”single”-task model**, as PCN is intended for scenarios where the ***number of channels is unknown***. As shown in **Table 3**, PCN is tailored for **zero-shot scenarios** and does not allocate parameters per channel but rather shares them across channels. For example, only PCN is applied in **Table 4**, as CN and ACN are not applicable to zero-shot scenarios.
> **L324–326 (Right column)** We attribute this to the fact that, unlike CN and ACN which assign each channel a distinct parameter, PCN assigns each prototype a distinct parameter,
>
> &nbsp;
>
> **[CN vs. ACN]**
> If your concern is the comparison between CN and ACN, ACN extends CN by incorporating local parameters that **dynamically** adapt based on the input TS, considering its dynamic nature (**L20--23 (Left column)**). As seen in **Table 8**, this enhancement improves performance while introducing only a negligible increase in computational complexity.
>
> &nbsp;
>
> **[Others]**
> Alternatively, if your inquiry pertains to any of the following, we provide the relevant information:
> - a) **Performance results**:
>   - [CN/ACN] **Tables 2, 6, 7**
>   - [PCN] **Tables 4, 5**
> - b) **Interpretation of results**: **Section 5.1**
> - c) **Experimental details**:
>   - [Datasets] **Appendix A.1**
>   - [Settings] **Appendix A.2**
>
> If this response does not fully address your concern, we would appreciate any further clarification or specific details you could provide.
>
> &nbsp;
>
> &nbsp;
>
> ## Question 2. Initialization of parameters of CN, ACN, PCN
> > Reviewer: *How do authors initialize the parameters of CN, ACN, PCN? How do the different initialization methods affect the metrics?*
>
> The initialization of the parameters for CN, ACN, and PCN is designed to ensure that ***no normalization occurs when learning has not yet taken place***:
> - The **scale** parameter ($\alpha$) is initialized to **1**.
> - The **shift** parameter ($\beta$) is initialized to **0**.
>
> This aligns with the default initialization used in **PyTorch normalization layers**:
> - (Pytorch) **Layer** normalization initialization: https://github.com/pytorch/pytorch/blob/1eba9b3aa3c43f86f4a2c807ac8e12c4a7767340/torch/nn/modules/normalization.py#L210
> - (Pytorch) **Batch** normalization initialization: https://github.com/pytorch/pytorch/blob/1eba9b3aa3c43f86f4a2c807ac8e12c4a7767340/torch/nn/modules/batchnorm.py#L93
>
> Therefore, the parameters for CN, ACN, and PCN are initialized as follows:
>
> - CN: $\alpha=1, \beta=0$
>
> - ACN: $\alpha^{\text{G}}=1, \alpha^{\text{L}}=0, \beta^{\text{G}}=1, \beta^{\text{L}}=0$, so that $\alpha=1, \beta=0$
>
> - PCN: $\alpha^{\text{P}}=1, \beta^{\text{P}}=0$
>
> This initialization is documented in the Python files within `/layers` in the attached anonymous GitHub link.
>
> We acknowledge that the explanation of initialization was omitted in the paper. As the reviewer pointed out, **we will include this explanation in the revised version**. Thank you for your feedback.
>
> &nbsp;
>
> Additionally, after testing on four ETT datasets and running experiments three times with different random initialization seeds (using PyTorch's default `nn.Parameter` initialization), **the effect on the average across four horizons was negligible**, differing only at the fourth decimal place.
>
> &nbsp;
>
> &nbsp;
>
> **If there are any unresolved issues, please feel free to discuss them with us!**

---

> > ### Comment · Reviewer_PycD · 2025-04-02
> >
> > - My expression is ambiguous. What I want to know is the **consumption of computation**. For example, I am more concerned about time complexity, space complexity, and running time.
> > - I can't get this detail from the article, but the rest of the paper is splendid. I will raise the score to 5 after the author adds the corresponding detail.

---

> > > ### Author Response · Authors · 2025-04-03
> > >
> > > Thank you for continuing to engage in the discussion.
> > >
> > > Below we provide a computational analysis of CN, ACN, and PCN (compared to LN) from the following **three perspectives**:
> > > 1. **Time Complexity**
> > > 2. **Space Complexity**
> > > 3. **Running Time**
> > >
> > > &nbsp;
> > >
> > > ## Notation
> > > $x \in \mathbb{R}^{B \times C \times D}$, where $K$ is the number of (predefined) clusters for PCN, $K < C$ in our experiments (as we set $K=5$ for all single-task settings (as referred in **L320–321 (Left column)**) and $K=20$ for TSFMs (as referred in **L418–419 (Right column)**).
> > >
> > > $D$ is set to 256 or 512 following the setting of each backbone in previous works, while $C$ varies across datasets, as detailed in **Table A.1**.
> > >
> > > &nbsp;
> > >
> > > ## 1. Time Complexity
> > > All methods (LN, CN, ACN, and PCN) share the basic four steps  (1. calculating the mean, 2. calculating the variance, 3. normalization, and 4. affine transformation), while ACN and PCN require an additional step.
> > >
> > > Among the basic four steps, steps 1--3 are **identical** across all methods, and step 4 is different among these methods; however, since they all employ element-wise multiplication, the time complexity of step 4 **remains unchanged**. Thus, **LN and CN have the same time complexity**, as CN follows the same computational steps as LN without any additional operations.
> > > For ACN and PCN, we additionally compute the weighted average of the parameters used in the affine transformation in step 4, **incurring additional time complexity**. Specifically, ACN uses weights based on channel similarities, resulting in additional $\mathcal{O}(C^2 D)$ time, while PCN uses weights based on channel-prototype similarities, resulting in additional $\mathcal{O}(CDK)$ time.
> > >
> > > The resulting time complexities are as follows:
> > > - LN: $\mathcal{O}(CD)$
> > > - CN: $\mathcal{O}(CD)$
> > > - ACN: $\mathcal{O}(C^2 D)$
> > > - PCN: $\mathcal{O}(CDK)$
> > >
> > > &nbsp;
> > >
> > > ## 2. Space Complexity
> > > Before delving into the details, we note that the proposed normalization methods can easily be implemented by **changing/adding a few lines of code for LN**. For example, CN can be implemented by simply applying the following change (similar change to beta as well):
> > > - LN: `alpha = nn.Parameter(torch.ones(D))`
> > > - CN: `alpha = nn.Parameter(torch.ones(C, D))`
> > >
> > > Unlike LN, our proposed method maintains learnable parameters **for each channel (or cluster)**, resulting in the following space complexity for the affine transformation parameters:
> > > - LN: $\mathcal{O}(D)$
> > > - CN: $\mathcal{O}(CD)$
> > > - ACN: $\mathcal{O}(CD)$
> > > - PCN: $\mathcal{O}(KD)$
> > >
> > >
> > > &nbsp;
> > >
> > > ## 3. Running Time (Training & Inference Time)
> > > The comparison of LN, CN, and ACN is provided in **Table 8**. Since PCN is missing in that table, we summarize the results together with PCN below, where the original iTransformer setting corresponds to iTransformer+LN.
> > > |iTransformer |+LN|+CN|+ACN|+PCN|
> > > |-|-|-|-|-|
> > > |Training time (sec/epoch)|7.7| 7.8|10.8|11.1|
> > > |Inference time (ms)|2.0|2.1|2.5|2.7|
> > > |Avg.MSE|.254|.159|.153|.176|
> > >
> > > If you are wondering why PCN underperforms compared to CN/ACN, please refer to **Question 1** in our initial rebuttal above.
> > >
> > > &nbsp;
> > >
> > > Again, **we sincerely appreciate your valuable feedback**. We will incorporate these points into the revision.

---

### Official Review · Reviewer_ZmeZ · 2025-03-22

**Overall Recommendation:** 3

**Summary:**

The authors propose a new method to adaptively normalize each time series channel distinctly through learned channel specific adaptive parameters. These adaptive parameters for each channel are data dependent are computed through a dynamic weighted summation of a similarity matrix computed between channel token embeddings.
  This  normalization allows models  to incorporate channel identity information. This allows different channels to produce different outputs even when provided the same input data to forecast on
Equipped with this adaptive normalization,  models can incorporate channel information when forecasting future values.


The authors also propose a variation of this normalization method that could be deployed for scenarios where the number of channels can change or are unknown (which is the case for time series foundation models). This can be done by reformulating the channel adaptive normalization parameters as the weighted sum of a learnable set of prototypes.

A major contribution of the proposed method is that this normalization scheme can be applied to any existing time series forecasting method.
The authors test their adaptive normalization scheme with various time series models (transformers, MLPs, Mamba). These include models which are already incorporate channel identification and models do not. When tested on different time series forecasting datasets,  the results show that the adaptive normalization improves forecasting performance for all models, even models that already incorporate channel identify information .

The authors also show how their proposed prototype based adaptive channel normalization scheme helps improve performance for Time Series foundation models.

**Claims And Evidence:**

The claims are supported by extensive experiments on different types of datasets and time series models. The improvement in performance over no adaptive normalization schemes suggest that the proposed scheme does enhance performance.


The authors also claim that the learned  normalization enhances feature representation and improves uniqueness of channel representation. This is backed by their experiments which show how the proposed method increases the entropy associated with representations across channels.

**Essential References Not Discussed:**

None that I can think of

**Experimental Designs Or Analyses:**

Yes, the soundness of the validity and experimental design makes sense.

The datasets used, and the baselines evaluated with all make sense. Particularly for results in Table2 and Table 3.


I checked the validity for experiments that showed how normalization learns feature representations which are more diverse/unique across channels.

**Methods And Evaluation Criteria:**

Yes, the proposed method and the evaluation criteria make sense for the model.

The method is compared with other normalization schemes, with various time series forecasting models on a variety of datasets

**Other Comments Or Suggestions:**

It would be helpful for readers to provide what different legend in different figures represent.

For example, LN in Figure 5? Is that layer normalization? This needs to be clarified

**Other Strengths And Weaknesses:**

I think the main contribution of the paper in terms of its methodology is strong, but a major weakness is the lack of clarity in the paper which hampers reading. I will gave examples of this in my comments/suggestions/questions

This is the main reason I would incline towards giving it a weak acceptance..

**Questions For Authors:**

- The description in Figure 1 is very confusing. The outputs are supposed to be yellow, and the inputs are supposed to be green. no?
- Why is the entropy values negative in Figure 5?
- Is $\hat{\alpha}_{b,c}^L$ a value or a vector in $d$? ? what dimension does it lie in (For equation 6) .
- What are $B$, $D$ in Algorithm 1 , text before equation4 . This is never clarified which makes things  confusing for the end user

**Relation To Broader Scientific Literature:**

The authors appropriately relate their finding to broader scientific literature. They put their work in context with models that incorporate channel identification  and those that not. They also put their work in context with commonly used normalization schemes or incorporate channel specific parameters (such as channel specific identified or Channel specific LORA)

**Theoretical Claims:**

The main paper doesn't have e theoretical claim.

---

> ### Author Rebuttal · Authors · 2025-03-31
>
> ## Suggestions 1. Explanation of Legends in Figures
> > Reviewer: *"It would be helpful for readers to provide what different legend in different figures represent. For example, LN in Figure 5? Is that layer normalization? This needs to be clarified*
>
> Thank you for your feedback. Due to **space limitations**, commonly used terms such as Channel Normalization (CN) and Layer Normalization (LN) **were not explicitly labeled** in the legends. To ensure clearer understanding, **we will add these clarifications in the revised version**.
>
> &nbsp;
>
> &nbsp;
>
> ## Question 1. [Figure 1] Confusing descriptions & Miscolored
> > Reviewer: *The description in Figure 1 is very confusing. The outputs are supposed to be yellow, and the inputs are supposed to be green. no?*
>
> **[Confusing descriptions]**
>
> Regarding the description in the figure, **"Figure 1: Channel Identifiability"** is intended to motivate the **necessity of CID**.
>
> To illustrate this, we present two cases: **“with CID”** and **“without CID”**. Specifically, in the **left** panel, when the local inputs are identical (green), non-CID models fail to distinguish between channels, producing identical outputs (yellow). In contrast, in the **right** panel, applying our proposed CN introduces CID, such that even if the local inputs are the same (green), CID models distinguish between channels, producing different outputs (yellow).
>
> We **will revise and clarify them** in the updated version. If there are still any confusing aspects regarding the figure, please feel free to ask us; we are happy to provide additional clarification.
>
> &nbsp;
>
> **[Miscolored]**
>
> Regarding the **coloring mistake**, we will fix the colors (green & yellow) in the revised version. Thank you for pointing that out!
>
>
> &nbsp;
>
> &nbsp;
>
> ## Question 2. Negative (approximated) entropy
> > Reviewer: *Why is the entropy values negative in Figure 5?*
>
> As the Gaussian entropy is ***approximated*** based on the data samples, the negative entropy values in **Figure 5** emerge. Previous studies ([1]--[5]) have also adopted this approximation (as mentioned in **L369--374**), and, as seen in [5] Sequence Complementor (AAAI 2025), negative values are observed in **Figure 2-(c)**, **Figure 5**, and **Figure S.6**.
>
> &nbsp;
>
> [1] Ma, et al. "Segmentation of multivariate mixed data via lossy data coding and compression." TPAMI (2007)
>
> [2] Yu, et al. "Learning diverse and discriminative representations via the principle of maximal coding rate reduction." NeurIPS (2020)
>
> [3] Chen, et al.  "Learning on Bandwidth Constrained Multi-Source Data with MIMOinspired DPP MAP Inference." IEEE Transactions on Machine Learning in Communications and Networking (2024)
>
> [4] Chen, et al. "Rd-dpp: Rate-distortion theory meets determinantal point process to diversify learning data samples." WACV (2025)
>
> [5] Chen, et al. "Sequence Complementor: Complementing Transformers For Time Series Forecasting with Learnable Sequences." AAAI (2025)
>
> &nbsp;
>
> &nbsp;
>
> ## Question 3. Is $\hat{\alpha}_{b, c}^{L}$, a vector or a scalar?
> > Reviewer: *Is $\hat{\alpha}_{b, c}^{L}$ a value or a vector in $d$? what dimension does it lie in (For equation 6)?*
>
>
>
>
> In the **Equation (6)**,
>
> - $\hat{S}_{b, c, i}$ is a scalar and
>
> - $\alpha_i^{\mathrm{L}}$  (or $\alpha_{i,:}^{\mathrm{L}}$)  is a $D$-dimensional vector,
>
> resulting in a $D$-dimensional vector.
>
> Thanks to the reviewer’s feedback, we found typos in **line 6 of Algorithm 2,3**: $\alpha_{b,c,d}$ should be the vector $\alpha_{b,c,:}$, not a scalar, and this also applies to the bias term $\beta_{b,c,d}$ in line 7.
>
> We agree that this might be difficult to understand with the explanation around **Algorithm 2**, we will provide a more detailed description of the dimensions in the main text and correct the error.
>
> &nbsp;
>
> &nbsp;
>
> ## **Question 4. Missing notations**
> > Reviewer: *What are $B$, $D$  in Algorithm 1, text before equation 4. This is never clarified which makes things confusing for the end user*
>
> Thank you for pointing that out. We found some of these definitions were missing. $B$ and $D$ represent the **"batch size"** and the **"number of channels"**, respectively.
>
> To improve clarity, we will explicitly define them in **Section 4** and also include them in the algorithm pseudocode.
>
> &nbsp;
>
> &nbsp;
>
> **If there are any unresolved issues, please feel free to discuss them with us!**

---

> > ### Comment · Reviewer_ZmeZ · 2025-04-05
> >
> > Thank you for providing answers to my questions. These changes would help improve clarity of the proposed work.

---

### Decision · Program_Chairs · 2025-05-01

**Decision:**

Accept (poster)

**Comment:**

This paper proposes Channel Normalization (CN) and its variants—Adaptive CN (ACN) and Prototypical CN (PCN)—to improve Channel Identifiability (CID) in time series models. CN assigns distinct affine parameters to each channel; ACN adapts these based on input, and PCN enables generalization to unknown or varying numbers of channels. The methods are lightweight, architecture-agnostic, and show strong empirical performance across diverse models and datasets.

The reviewers appreciated the broad applicability, consistent improvements, and simplicity of the approach. While some raised clarity concerns and questioned PCN’s CID ability, the authors addressed these through a detailed and thoughtful rebuttal, clarifying that PCN offers cluster-level identification and is tailored for foundation model settings.

Overall, this is a well-motivated and practically useful contribution to time series modeling. The revisions and response fully resolve remaining concerns. I recommend acceptance.